# The bone ecosystem facilitates multiple myeloma relapse and the evolution of heterogeneous drug resistant disease

Ryan T. Bishop [1,6], Anna K. Miller [2,6], Matthew Froid[2,3], Niveditha Nerlakanti [1,3], Tao Li[1], Jeremy S. Frieling [1], Mostafa M. Nasr[1,3], Karl J. Nyman[1,3], Praneeth R. Sudalagunta [4], Rafael R. Canevarolo [4], Ariosto Siqueira Silva[4], Kenneth H. Shain[1,5], Conor C. Lynch [1,6] ✉ & David Basanta [2,6] ✉

Multiple myeloma (MM) is an osteolytic malignancy that is incurable due to the emergence of treatment resistant disease. Defining how, when and where myeloma cell intrinsic and extrinsic bone microenvironmental mechanisms cause relapse is challenging with current biological approaches. Here, we report a biology-driven spatiotemporal hybrid agent-based model of the MM-bone microenvironment. Results indicate MM intrinsic mechanisms drive the evolution of treatment resistant disease but that the protective effects of bone microenvironment mediated drug resistance (EMDR) significantly enhances the probability and heterogeneity of resistant clones arising under treatment. Further, the model predicts that targeting of EMDR deepens therapy response by eliminating sensitive clones proximal to stroma and bone, a finding supported by in vivo studies. Altogether, our model allows for the study of MM clonal evolution over time in the bone microenvironment and will be beneficial for optimizing treatment efficacy so as to significantly delay disease relapse.

Multiple myeloma (MM) is the second most common hematological malignancy in the US[1], characterized by the clonal expansion of plasma cells localized primarily in the bone marrow microenvironment of the skeleton[2,3]. In the bone microenvironment, MM cells disrupt the finely tuned balance of trabecular and cortical bone remodeling to generate osteolytic lesions. This is primarily via MM-induced inhibition of osteoblastic bone formation and increased osteoclastic bone resorption. There are currently several approved therapies to treat MM including, but not limited to, proteasome inhibitors (e.g., bortezomib and carfilzomib), chemotherapy (e.g., melphalan), immunomodulatory drugs (e.g., lenalidomide), and cellular and non-cellular immunotherapies (e.g., B cell maturation antigen (BCMA) chimeric antigen receptor- (CAR) T cells and daratumumab) that significantly contribute to the long-term survival of MM patients[4]. However, MM remains fatal as nearly all patients ultimately become refractory to various lines of therapy. A greater understanding of how resistant MM emerges in the clinical setting can yield new treatment strategies to slow relapse times and extend the efficacy of standard of care therapies.

In MM and other cancers, the tumor microenvironment is a well-known contributor to the development of treatment resistance[5–10]. MM is an archetype of cancer cell-stroma interactions whereby MM cells reside and metastasize systemically within the bone marrow microenvironment[11]. In the bone ecosystem, MM interacts closely with

[1]Department of Tumor Microenvironment and Metastasis, H. Lee Moffitt Cancer Center and Research Institute, Tampa, FL 33612, USA. [2]Department of Integrated Mathematical Oncology, H. Lee Moffitt Cancer Center and Research Institute, Tampa, FL 33612, USA. [3]The Cancer Biology Ph.D. Program, University of South Florida, Tampa, FL, USA. [4]Department of Metabolism and Physiology, H. Lee Moffitt Cancer Center and Research Institute, Tampa, FL 33612, USA. [5]Department of Malignant Hematology, H. Lee Moffitt Cancer Center and Research Institute, Tampa, FL 33612, USA. [6]These authors contributed equally: Ryan T. Bishop, Anna K. Miller, Conor C. Lynch, David Basanta. ✉e-mail: conor.lynch@moffitt.org; david@cancerevo.org

mesenchymal stem cells (MSCs) and other progenitor cells[6,12–15] that secrete MM pro-survival and proliferative factors. Additional studies have shown that the bone marrow microenvironment can also protect MM cells from applied therapy, a phenomenon known as environment mediated drug resistance (EMDR)[7,8,16]. Reciprocally, it is well established that MM-derived factors stimulate osteoblast lineage cells to recruit and activate osteoclasts through receptor activator of nuclear factor kappa-B ligand (RANKL) production. Osteoclasts degrade the bone matrix to release stored cytokines and growth factors (e.g., transforming growth factor-beta, TGF-β) that, in turn, support MM survival and growth[2,3,5–10]. MM cells also suppress osteoblast maturation, thus tipping the balance towards excessive bone destruction[2,3]. This vicious cycle leads to complications for MM patients such as hypercalcemia and pathological fracture that greatly contribute to morbidity and mortality[17].

While EMDR can contribute to drug resistance, MM intrinsic mechanisms, such as genetic and epigenetic alterations, also play a significant role in the process. For example, increased expression of the anti-apoptotic BCL2 protein family is important in mediating MM resistance to proteasome inhibition[18–21]. In fact, depending on selective pressures by different lines of therapies and clonal mutation capacity, MM may evolve to develop multiple mechanisms of resistance[5,22,23]. While EMDR and intrinsic drug resistance mechanisms contribute to the emergence of refractory disease, several questions remain: (i) How does the interplay between each mechanism contribute to the evolution of resistant MM? (ii) How does each mechanism contribute to the heterogeneity of the disease under either control or treatment conditions? (iii) Could targeting EMDR alter the course of the disease or its responsiveness to applied therapy thereby yielding significantly greater depths of treatment response? In the context of the bone marrow microenvironment and the direct/indirect communications occurring in a temporal and spatial fashion between MM and stromal cells, these questions remain difficult to address using current in vivo and ex vivo approaches.

Mathematical models of the tumor microenvironment have become a valuable approach that allows us to interrogate the interactions between multiple cellular populations over time under normal or disease conditions. In particular, hybrid cellular automata (HCA) models which couple discrete cell-based models with partial differential equations include spatial aspects enabling users to locate where phenomena are occurring. Further, mathematical models have the potential for clinical translatability and with several already having been used to study the impact of treatment on the tumor microenvironment and inform clinical decision making to delay resistant disease in patients[24–31]. We therefore leveraged these modeling approaches to examine MM progression and spatial evolution in response to therapy.

In this work, we generated an HCA model powered by biological parameters from a mouse model of MM and demonstrate how it captures both normal bone turnover, and spatiotemporal cellular dynamics during MM progression in bone, such as loss of osteoblasts, increased stromal cell infiltration, osteoclast formation, and osteolysis. Additionally, we incorporate the proliferative and protective bone marrow microenvironment effects on MM mediated by osteoclastic bone resorption and bone marrow stromal cells. In the treatment setting, the HCA demonstrates that EMDR contributes to minimal residual disease, protecting tumors from complete eradication. Moreover, this reservoir of MM cells increases the likelihood of unique resistance mechanisms arising over time and greatly contributes to the evolution of MM heterogeneity compared to non-EMDR scenarios. These results highlight the importance of the bone marrow microenvironment in contributing to MM resistance and patient relapse and provide a strong rationale for targeting both MM and bone marrow microenvironment mechanisms for optimal treatment response.

## Results

### The HCA model recapitulates the key steps of trabecular bone remodeling

Trabecular bone remodeling is a continuously occurring process that is essential for regulating calcium homeostasis and bone injury repair (Fig. 1a)[32,33]. This process occurs throughout the skeleton at individual sites known as basic multicellular units (BMUs). The BMU in our HCA consists of five different cell types, including precursor osteoclasts, active osteoclasts, mesenchymal stem cells (MSCs), preosteoblasts, and osteoblasts. Additionally, we include central signaling molecules such as RANKL, a driver of preosteoclast motility and OC formation[34] and osteoclastic release of bone derived factors (BDFs) such as TGF-β of which bone is a major reservoir[35]. In the HCA, we use TGF-β as a representative BDF given its known biphasic effects on cell behavior and its concentration dependent ability to modulate preosteoblast proliferation and maturation to bone forming osteoblasts[35–39]. We defined the roles and interactions of these key cellular populations and the factors that govern their behavior during the remodeling process using parameters derived experimentally or from the literature (Fig. 1b, c, Supp Fig. 1a and Supplementary Tables 1–6). It is important to note that each cell in the HCA behaves as an autonomous agent that can respond to surrounding environmental cues independently. In our HCA model, new bone remodeling events are initiated over time at randomly selected locations such that multiple BMUs may be present at a given time as opposed to our previous HCA model[27,40–42]. We demonstrated that each BMU undergoes five key phases of bone remodeling: initiation, resorption, reversal, formation, and quiescence[43] ultimately returning to homeostasis (Fig. 2 and Supplementary Movie 1). We also observed that each simulation ($n = 25$) generated multiple BMUs over time, resulting in the replacement of the majority of the bone over a four-year period (Fig. 2, Supplementary Movie 1, Supplementary Fig. 1). We also noted that while the bone area to total area (BA/TA) remains relatively constant the shape of the trabecular bone changes, underscoring the dynamic nature of HCA model and how it reflects the in vivo scenario.

### The HCA model captures dose-dependent effects of BDF and RANKL

To challenge the robustness of our bone remodeling HCA, we altered the levels of key factors controlling bone homeostasis. Our in vitro experimental data show that neutralizing TGF-β or addition of exogenous TGF-β in osteoblastic cell cultures increases and decreases mineralization respectively, a finding that is consistent with the literature[35,39,44] (Supplementary Fig. 2a). Similarly, lowering TGF-β levels in silico led to a dramatic increase in bone volume (Supplementary Fig. 2b–e). Interestingly, the HCA results showed that when TGF-β levels dropped below 90%, there was significant bone loss (Supplementary Fig. 2b–e) due to abrogation of MSC proliferation leading to reduced numbers of preosteoblasts that contribute to bone formation (Supplementary Fig. 2e). These data are reflective of the known biphasic roles of TGF-β on cell behaviors and processes at high and low concentrations[35,39,44]. In addition, we demonstrated the sensitivity of OC formation in the HCA model in response to changes in RANKL levels (Supplementary Fig. 2f–h). These data demonstrate that the complex HCA model can integrate the behaviors and responses of multiple highly coupled cell types and stimuli to achieve responses reflective of known biology.

### The HCA model integrates the impact of standard of care therapies on bone remodeling

Next, we assessed the response of the HCA model to standard of care therapies used for the treatment of MM such as zoledronate (ZOL) and bortezomib (BTZ). These compounds can block osteoclast function and formation respectively but, as we and others have shown, they can also stimulate osteoblast mediated bone formation[45–48]

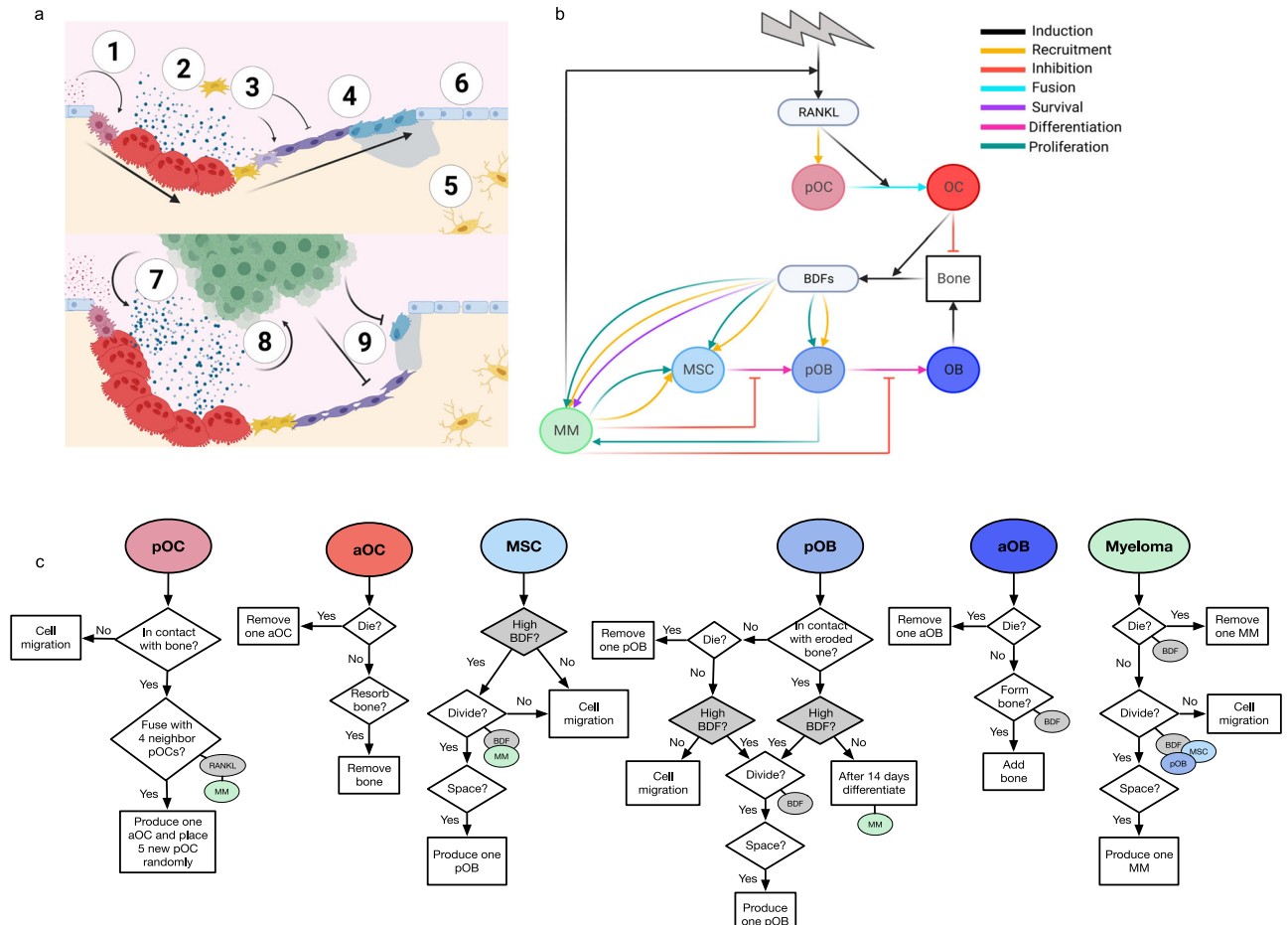

**Fig. 1 | Development of a hybrid HCA of the naïve and myeloma bone microenvironment. a** Bone-lining osteoblast lineage cells release RANKL inducing fusion and maturation of osteoclasts (1). Osteoclasts resorb the bone, releasing stored bone-derived factors (BDFs) such as TGF-β (2). TGF-β recruits local MSCs and stimulates asymmetric division in to preosteoblasts (2). When TGF-β levels remain high, preosteoblasts rapidly proliferate. Following osteoclast apoptosis, release of TGF-β falls and preosteoblasts differentiate to mature bone producing osteoblasts (3). While TGF-β levels remain low, osteoblasts produce new bone (4). As bone returns to normal, a fraction of the osteoblasts is buried within the matrix becoming terminally differentiated osteoblasts (5), the remaining osteoblasts undergo apoptosis, or become quiescent bone-lining cells (6). Myeloma cells enhance the formation of osteoclasts (7), enhanced bone resorption produces higher levels of BDFs which fuel myeloma growth (8) and inhibit osteoblast differentiation (8) and activity (9). Created with biorender.com. **b** Interaction diagram between cell types in the HCA and factors such as BDFs and RANKL (created with biorender.com). A more detailed interaction diagram with references can be found in supplementary fig. 1. **c** Flowcharts describing the sequence of steps followed by preosteoclasts, osteoclasts, MSCs, preosteoblasts, osteoblasts, and myeloma cells.

(Supplementary Fig. 3a, b and Supplementary Fig. 4a, b). Our results show that, as expected, both ZOL and BTZ therapy increased bone volume over the treatment period by enhancing osteoblastic bone formation and blocking osteoclast activity/formation (Supplementary Fig. 3c–h, Supplementary Fig. 4c–h). While osteoclasts are impacted by both treatments, the HCA recapitulates their distinct mechanisms of action showing that ZOL targets actively resorbing osteoclasts as demonstrated by a reduction in bone resorption per OC (Supplementary Fig. 4h) while BTZ limits the fusion of osteoclast precursors as shown by a reduction in cumulative osteoclast numbers with treatment (Supplementary Fig. 3g). Collectively, these data demonstrate the ability of our HCA model to capture normal bone remodeling over extended periods and the appropriate response of the model to applied therapeutics used for the treatment of MM.

### The HCA model incorporates the MM-bone ecosystem vicious cycle

A key component of MM growth involves the initiation of a feed-forward vicious cycle leading to reduced bone formation and enhanced osteoclastic bone destruction (Fig. 1a). MM is also known to interact with several cellular components of the bone ecosystem such as MSCs[2,16]. To incorporate these effects and MM growth characteristics into the HCA, mice were inoculated with U266-GFP MM cells and at end point, bones were processed and stained for proliferation (Histone H3 phosphorylation (pHH3)) and apoptosis markers (Terminal deoxynucleotidyl transferase dUTP nick end labeling (TUNEL)). Our data show that the highest proportion of proliferating MM cells are located within 50μm of bone (Fig. 3a, b). We noted that fewer cells (TUNEL+) underwent apoptosis in this area (Fig. 3c, d) highlighting the protective effect of BDFs being generated from bone degradation. To consider the effect of bone marrow stromal cells on MM cells distal from bone, we performed in vitro experiments and found a significant increase in the proliferation of MM cells grown in the presence of conditioned medium from MSCs and preosteoblasts, but not osteoblasts (Fig. 3e, f). To integrate these observations into the HCA, we assume that the intrinsic rate of MM cell division increases with BDF and when a preosteoblast or MSC is near the MM cells (Fig. 3g, h and Supplementary Equation 11).

Next, to study the interactions of the MM-bone marrow microenvironment in the HCA model, we initialized each simulation with a single MM cell near a newly initiated bone remodeling event and the formation of an osteoclast. This assumption is a simplification of the

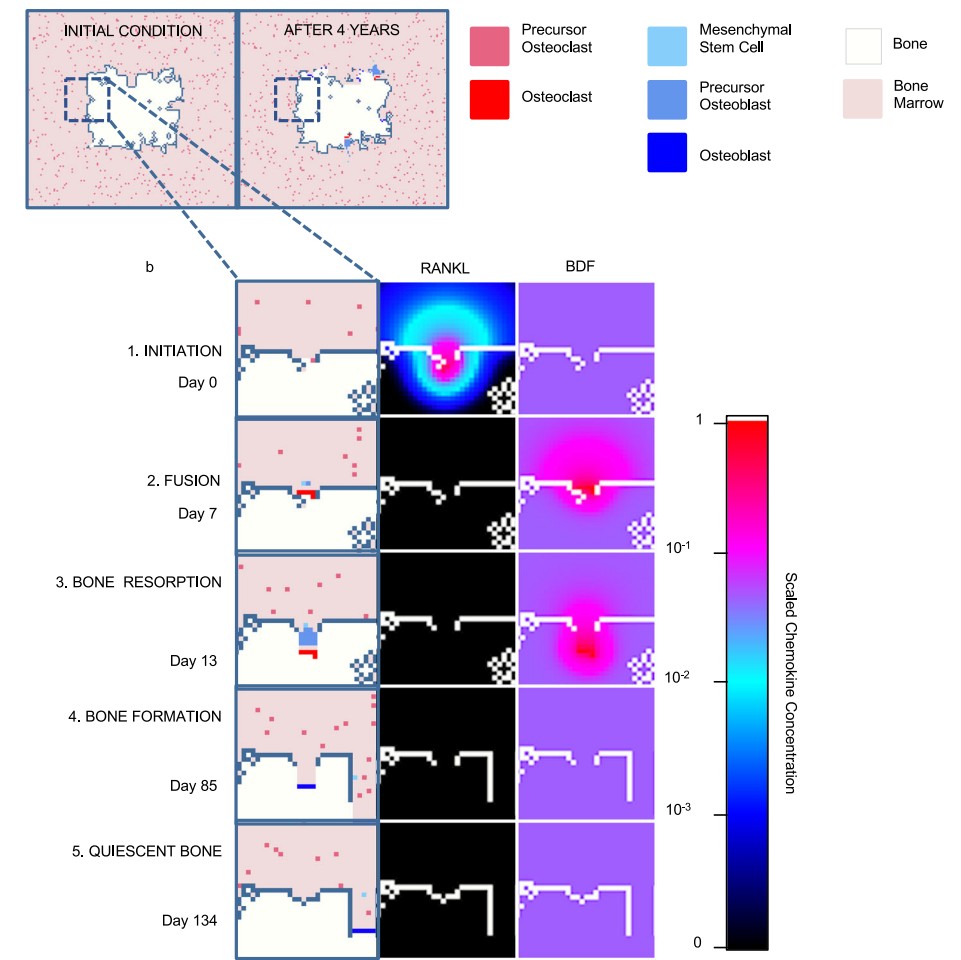

**Fig. 2 | The HCA model captures all stages of bone homeostasis. a** Images from simulations at initial conditions (left) and after 4 years (right). Legend depicts colors of cell types in the model. **b** BMU is initiated in response to release of RANKL from bone lining cells (1). Preosteoclasts migrate to RANKL and fuse to form an osteoclast under highest concentrations (2). Osteoclasts resorb bone and stored BDFs are released which recruits MSCs. MSCs divide asymmetrically producing preosteoblasts which proliferate rapidly under high TGF-β conditions. Upon completion of resorption, BDF levels fall allowing for preosteoblast differentiation to osteoblasts, osteoblasts form new bone (4) and ultimately undergo apoptosis or become quiescent bone lining cells (5).

colonization process, in which rare colonizing myeloma cells migrate to the endosteal niche[49,50]. The HCA model revealed exciting temporal dynamics where MM promotes successive phases of bone destruction mediated by osteoclasts that in turn drives MM growth, thereby successfully recapitulating the vicious cycle (Supplementary Fig. 4a and Supplementary Movie 2). As expected, the mathematical model demonstrated increasing tumor burden over time (Fig. 4a, b) with rapid loss of bone, correlating inversely with tumor burden (Fig. 4c, d). These findings are consistent with in vivo data obtained from the U266-GFP mouse model of MM which showed increasing levels of GFP positive cells in the bone marrow following ex vivo flow cytometric analysis (Fig. 4e) combined with a rapid onset of trabecular bone loss (BV/TV; Fig. 4f, g) and associated trabecular bone parameters as assessed by high resolution μCT (Supplementary Fig. 5). The similarities in trends between the in vivo model and the HCA outputs, indicated that the HCA model was accurately recapitulating the MM-bone vicious cycle.

Furthermore, the spatial and temporal nature of the HCA allowed us to make several observations: 1) In the model, the recruitment of MSCs in silico increases over time (Fig. 5a) as we and others have previously reported[51,52], 2) the model accurately shows a suppression of adult osteoblast formation (Fig. 5b) due to inhibition of differentiation by MM[2,53], and 3) the model predicts an early increase in preosteoblasts followed by a reduction in numbers (Fig. 5c), consistent

with previous reports[54], while there is a concomitant increase in osteoclasts (Fig. 5d)[2,53]. We also compared these population dynamics with our in vivo model. In tissue sections derived the U266-GFP model from various time points (days 10, 40 and 100), we measured MSC, preosteoblast, osteoblast, and osteoclast content and observed strong agreement with cell population trends generated by the MM HCA (Fig. 5e–h and Supplementary Fig. 6). Taken together, these data demonstrate the ability of the MM HCA to capture the vicious cycle of MM progression in the bone microenvironment over time.

**Environment-mediated drug resistance increases minimal residual disease and leads to higher relapse rates**

We and others have shown that BTZ demonstrates a dose dependent cytotoxic effect on MM (Supplementary Fig. 3a)[55,56]. The compound is also effective in vivo but MM still grows albeit at a significantly slower rate[55]. Patients also typically become refractory to BTZ. This indicates that MM cells are 1) protected by the surrounding bone ecosystem, or 2) due to selective pressure and mutation or other mechanisms such as epigenetic changes, drug resistant clones emerge, or 3) that both ecological and evolutionary contributions can occur in parallel. Here we sought to use the spatial nature of the HCA model to gain a deeper understanding of how MM resistance evolves focusing initially on the contribution of the bone ecosystem, i.e., EMDR. We confirmed the EMDR protective effect conferred by bone stromal cells on MM in the

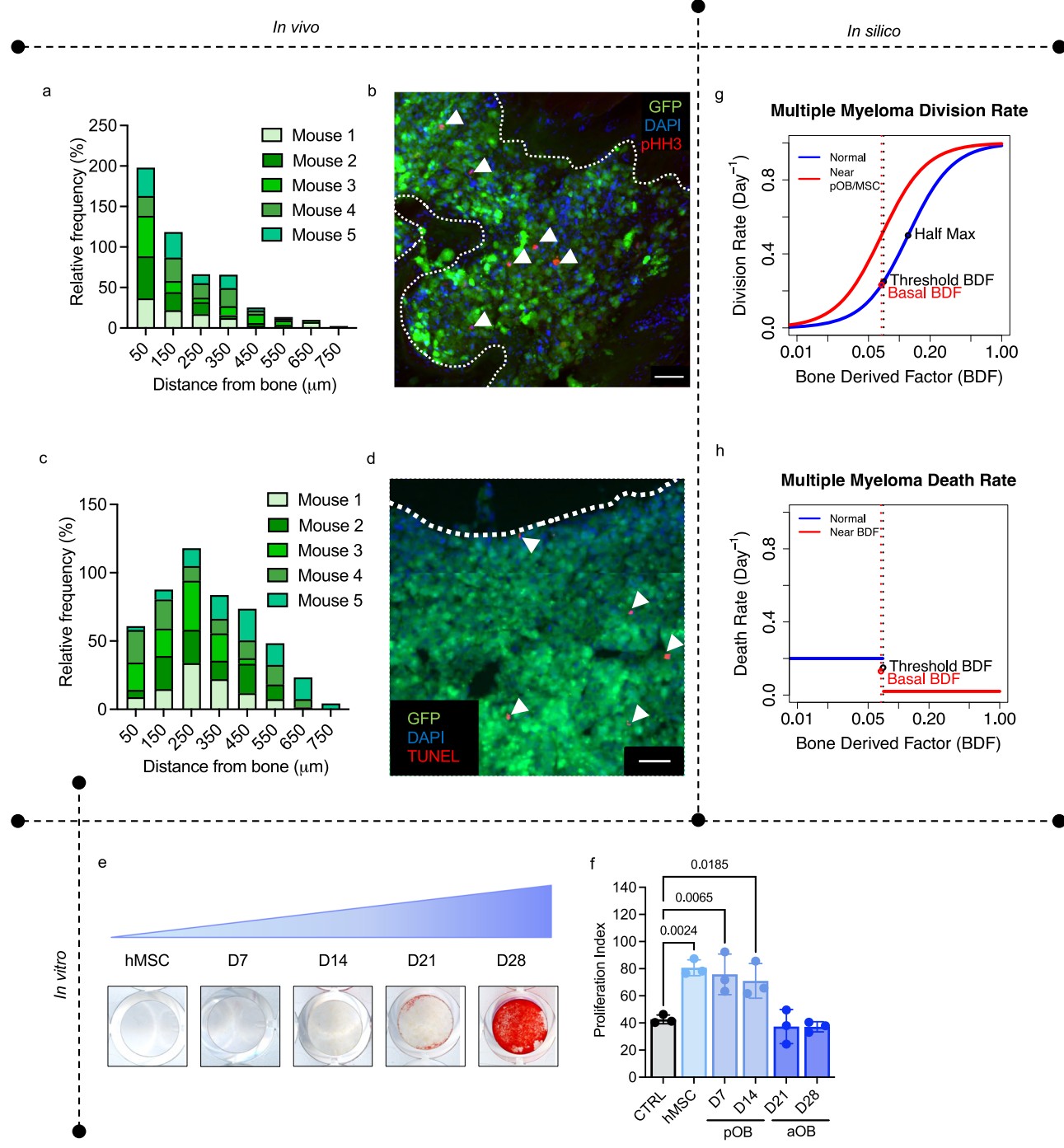

**Fig. 3 | Multiple myeloma cells receive proliferative and survival advantages from the bone marrow microenvironment. a** Quantification of the distance of phosphorylated histone H3 positive (pHH3+; red) myeloma cells (green) to the nearest trabecular or cortical bone in U266GFP-bearing mice 100 days post inoculation. $N = 5$ tibia. **b** Representative images from experiment in a, DAPI (blue) was used as a nuclear counterstain. White dotted line indicates tumor bone interface. White scale bar, 50 microns. **c** Quantification of the distance of TUNEL+ (red) myeloma cells (green) to the nearest trabecular or cortical bone in U266GFP-bearing mice 100 days post inoculation. $N = 5$ tibia. **d** Representative image from experiment described in c, DAPI (blue) was used as a nuclear counterstain. White dotted line indicates tumor bone interface. White scale bar, 50 microns. **e** Images of

huMSCs differentiated to different stages of the osteoblast lineage. Cells were stained with Alizarin Red to identify mineralization. **f** Mean proliferation index of CM-DiL stained U266 cells 7 days after growth in 50% conditioned medium from control wells or cells of the osteoblast lineage. Results are displayed as means of 3 independent biological replicates. Values are mean ± SD. **g** Plot of functional form used to represent the division rate of myeloma cells in the presence and absence of preosteoblasts/MSCs. **h** Plot of functional form used to represent the death of myeloma cells when BDF is above and below a threshold. Statistical significance was derived by ordinary one-way ANOVA with a Dunnett's test for multiple comparisons (**f**). Source data are provided as a Source Data file for **a**, **c** and **f**. Source data for **g** and **h** can be accessed at DOI 10.17605/OSF.IO/TNAX9.

presence of BTZ in vitro (Fig. 6a) and integrated this effect into the HCA model (Fig. 6b). BDF was also included since it can provide a survival advantage to MM cells in the presence or absence of treatment[57,58]. Next, we initialized HCA simulations with a homogeneous sensitive MM

population. BTZ was applied once the tumor burden reached a volume of 10% to mimic the scenario of diagnosis in the clinical setting. Our results show that without EMDR, BTZ completely eradicated the disease at high doses (Fig. 6d, Supplementary Fig. 3j), consistent with our

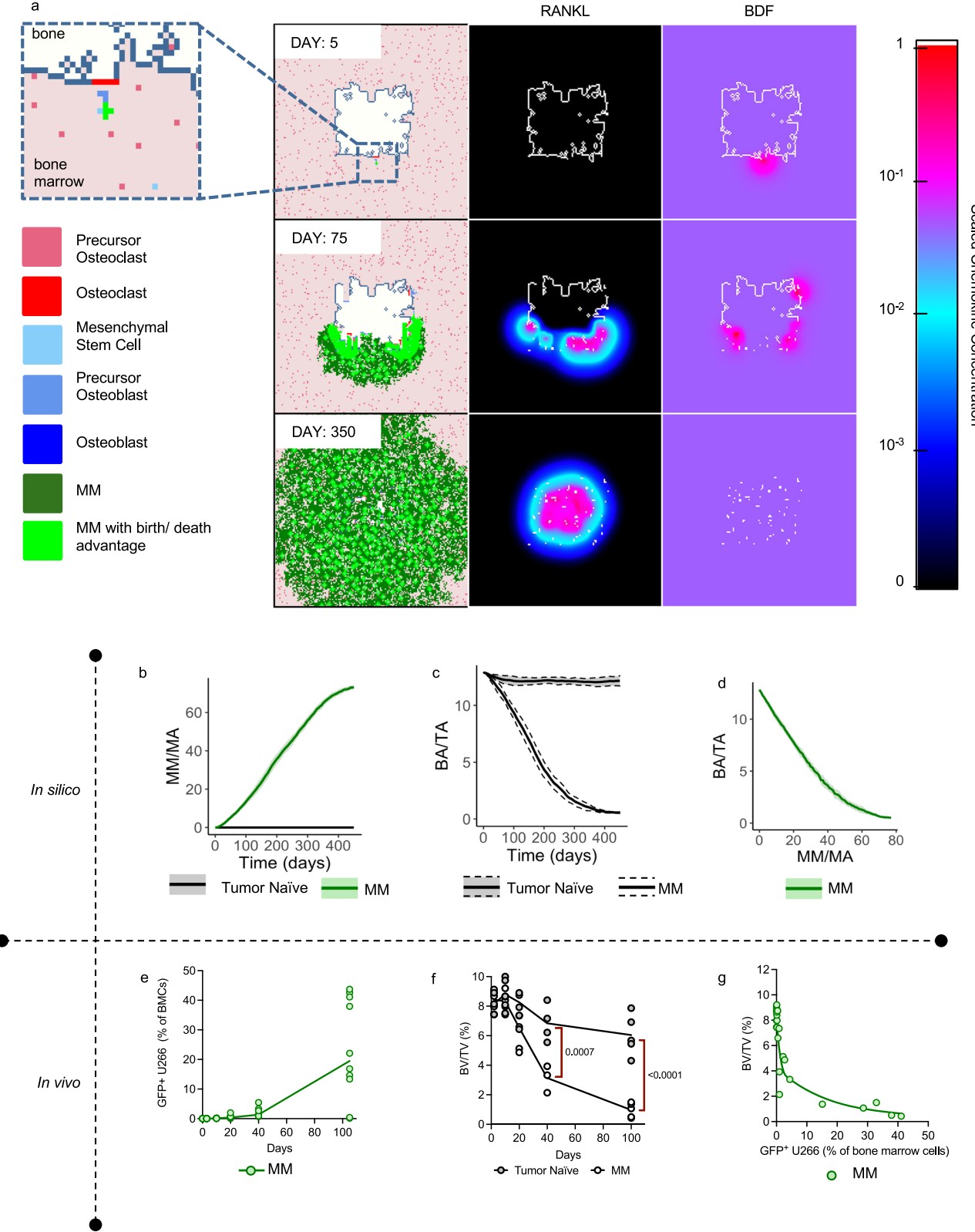

**Fig. 4 | Computational and biological model outputs of myeloma growth and bone dynamics. a** HCA model images showing single myeloma cell (Day 5), colonization of the marrow by MM cells (Day 75), increased osteoclastogenesis (Day 75, RANKL), bone resorption (Day 75; BDF) and eventual takeover of the marrow by MM cells (Day 350). Light green myeloma cells indicate MM cells with proliferative or survival advantage. **b** Myeloma growth dynamics in HCA model in the absence of treatment. Values are mean ± SD. **c** Myeloma induced loss of trabecular bone in HCA model compared to normal bone homeostasis. Values are mean ± SD. **d** Bone loss decreases rapidly with myeloma expansion in silico. Values are mean ± SD. **e** Mean Myeloma growth in bone marrow of mice inoculated with U266-GFP cells over time. $N = 3$–5 tibia per time point. **f** Trabecular bone volume fraction (BV/TV) was assessed ex vivo with high-resolution microCT. $N = 3$–5 tibia per time point. **g** Biological model shows rapid myeloma-induced bone loss. Statistical significance was derived by two-way ANOVA with a Šídák's test for multiple comparisons (**f**). Source data are provided as a Source Data file for **e**–**g**. Source data for **b**–**d** can be accessed at DOI 10.17605/OSF.IO/TNAX9.

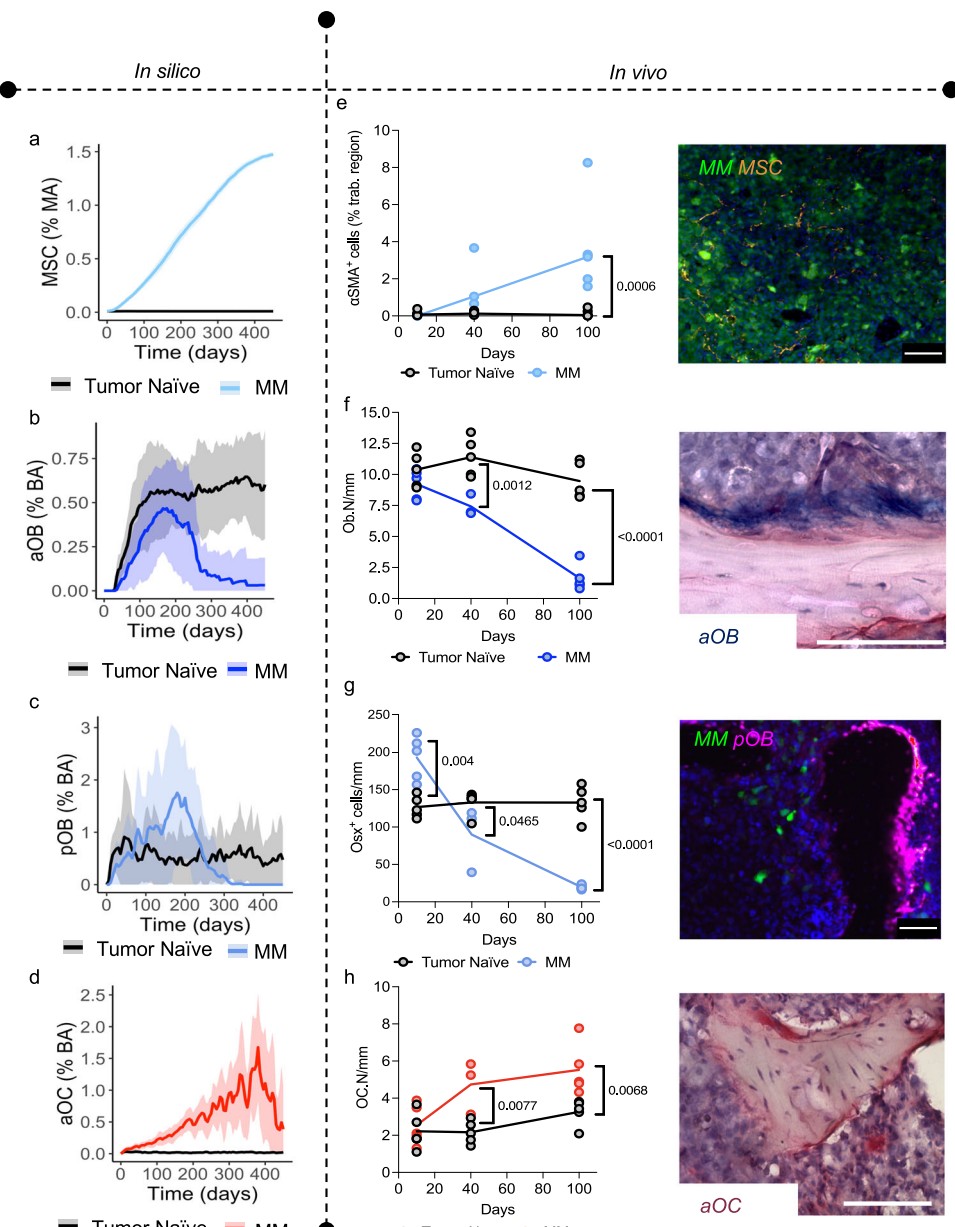

**Fig. 5 | Computational and biological model outputs of cell types in the myeloma bone microenvironment.** Computational model outputs of increasing MSC percentage of the marrow area (%MA) (**a**), loss of osteoblasts (**b**) rise and fall of preosteoblast percentage (**c**) and increasing osteoclast percentage due to growth of MM (**d**) Values are mean ± SD. Ex vivo analysis of histological sections from the U266-GFP myeloma model ($N = 3$-$5$ tibia per time point) demonstrates increasing presence of αSMA+ MSCs (**e**), loss of ALP+ cuboidal osteoblasts (**f**), early increase and subsequent reduction of OSX+ preosteoblasts (**g**) and increasing numbers of TRAcP+ multinucleated osteoclasts (**h**) compared to tumor naïve mice. Scale bars are 50 microns (**e**, **g**) 100 microns (**f**, **h**). Statistical significance was determined by two-way ANOVA with a Šídák's test for multiple comparisons (**e**–**h**). Source data are provided as a Source Data file for **e**–**h**. Source data for **a**–**d** can be accessed at DOI 10.17605/OSF.IO/TNAX9.

in vitro findings. However, when EMDR effects are included in the model, BTZ reduced MM burden significantly, but the disease persisted at a stable albeit lower level and did not grow over time compared to existing in vivo data (Fig. 6c-d). These data suggest that EMDR does not directly contribute to MM relapse but does protect a small volume of MM cells from BTZ. Therefore, to address the relapse and the emergence of BTZ resistance in MM cells, we integrated a drug resistance probability ($p_\Omega = 10^{-4}$ or $p_\Omega = 10^{-3}$) during cell division. These rates are estimated based on the current literature, in which mammalian cancer cells have a probability of developing resistance to numerous treatments of between $10^{-3}$ to $10^{-6}$ [59,60]. Based on our in vitro observations, we also incorporated a cost of resistance[61] in the model such that BTZ-resistant cells proliferate at a

slower rate and are out-competed by PI-sensitive cells in the absence of treatment (Supplementary Fig. 7).

We then repeated our *in-silico* experiments and observed that with the lower resistance probability ($p_\Omega = 10^{-4}$), EMDR increased the likelihood of tumor progression with more simulations likely to relapse (54.4%, $n = 125$) compared to tumors without EMDR (6.4%, $n = 125$; Fig. 6e and g; Supplementary Movie 3). Similarly, with a higher resistance probability ($p_\Omega = 10^{-3}$) tumors were more likely to relapse when EMDR is present (100%, $n = 25$) compared to tumors without EMDR (56%, $n = 25$, Fig. 6f and h; Supplementary Movie 4). However, the higher resistance probability also increased the likelihood of tumor progression regardless of EMDR, implying a potentially greater role for MM intrinsic resistance mechanisms during tumor relapse. To

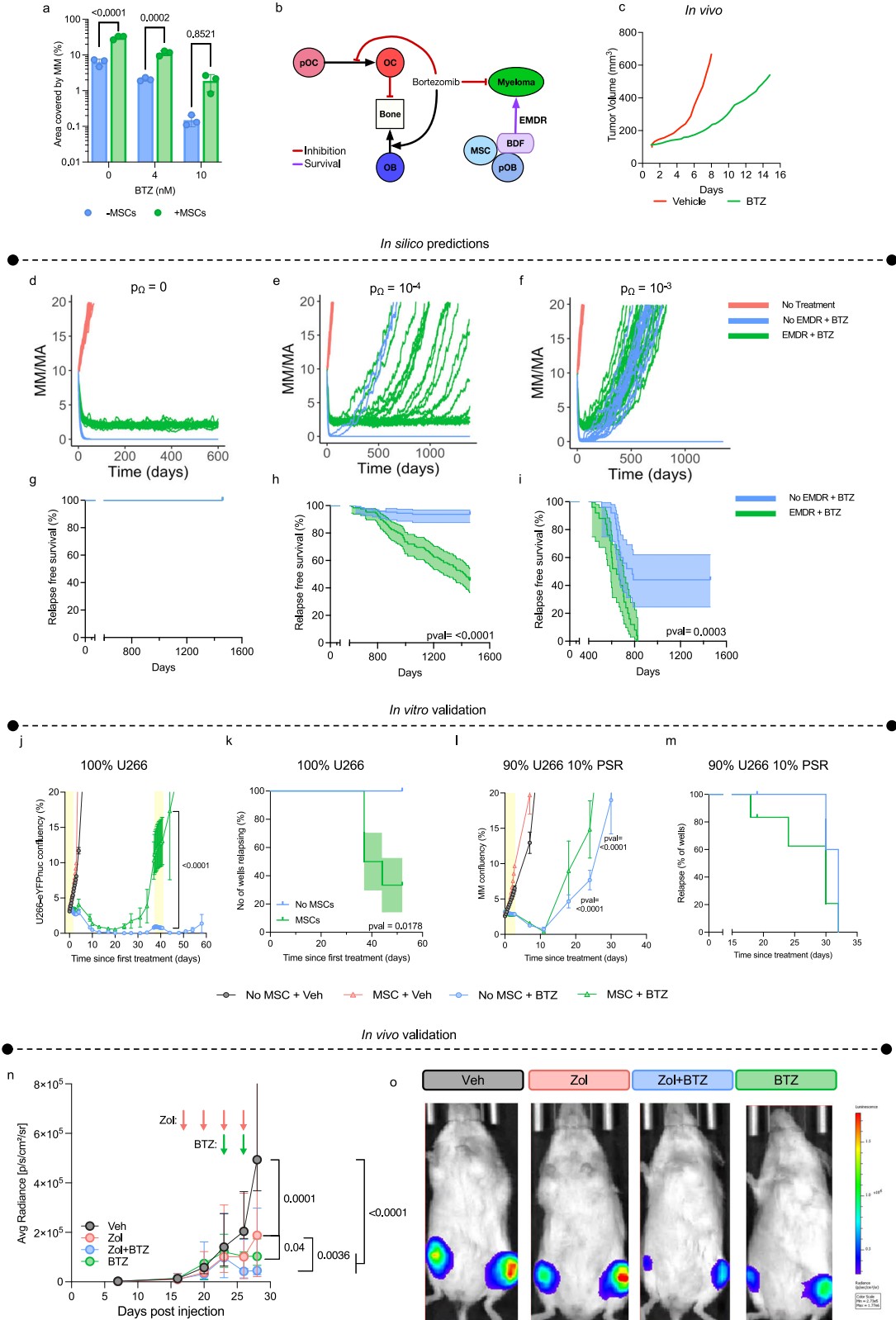

strengthen these findings, we tested a more clinically relevant treatment strategy, in which BTZ therapy was pulsed (2 weeks on, 1 week off) and observed similar results (Supplementary Fig. 8). To validate our in silico results, we utilized both in vitro and in vivo approaches. To this end, U266 MM cells were cultured alone (monocultures) or with protective human MSCs (huMSCs) and exposed to vehicle or BTZ. BTZ-treated U266 groups, responded to BTZ as evidenced by a reduction in

MM confluency (Fig. 6j). However, we observed that BTZ was less efficient when U266 were co-cultured with huMSC and could recover over a 30-day period, whereas monocultures did not. Further, co-cultured U266 cells treated with a second dose of BTZ, continued to proliferate, even beyond initial confluency, and thus were considered "relapsed", whereas this was not the case in monocultures (Fig. 6k). Since the propensity of MM cells to develop resistance cannot be

**Fig. 6 | EMDR contributes to minimal residual disease and relapse. a** GFP + U266 MM were plated alone or with huMSCs in the presence or absence of BTZ; MM burden was measured after 72 h by area covered by GFP+ cells. *n* = 3 biological independent samples. Values are mean ± SD. **b** Interaction diagram showing the cell types and factors in the HCA that are affected by BTZ or contribute to EMDR. **c** In vivo growth of MM cells with and without BTZ treatment, data was taken from[42]. Model outputs of MM growth with continuous BTZ treatment, +/− EMDR when MM cells do not develop resistance (**d**) or with a probability to develop resistance ($10^{-4}$, **e** or $10^{-3}$, **f**) and develop resistance. **g–i**, Kaplan-Meier plot of relapse-free survival from simulations described in **d–f**. Error bands represent 95% confidence intervals. **j** Nuclear-eYFP+ U266 were cultured alone or with huMSCs in the presence or absence of 10 nM of BTZ. Mean MM growth (*n* = 6 biologically independent samples per group) was tracked by eYFP over 60 days. Values are mean ± SD. Yellow shading indicates periods when treatment was on. **k** Kaplan-Meier plot of 'relapse' (wells reaching >20% MM confluency) from experiment described in **j**. **l** Mean MM cell numbers (initial seeding 90% U266- Nuclear-eYFP+ and 10% PSR-RFP+; *n* = 6

biologically independent samples per group) were cultured alone or with huMSCs in the presence or absence of 10 nM of BTZ. **m** Kaplan-Meier plot of 'relapse' (wells reaching >20% MM confluency) from experiment described in **l**. Values are mean ± SD. *n* = 6 biologically independent samples per group. *n* Median U266 growth by BLI after MM cells (90% U266-GFP⁺Luc⁺, 10% PSR-RFP) were tail vein injected into NSG mice. Values are median ± 95% confidence intervals. Mice were divided into two groups and pre-treated with vehicle or Zol (30 µg/kg) for 1 week prior to mice being randomized and treated with either vehicle (*n* = 5 mice), Zol (*n* = 5 mice), Zol +BTZ (*n* = 7 mice), or BTZ (0.5 mg/kg; *n* = 4 mice). Pink and green arrows indicate days of Zol or BTZ treatment respectively. o, Representative BLI IVIS images from the day 28 of the experiment described in *n*. Statistical significance was determined by two-way ANOVA with Bonferroni correction (**a**), Šídák's (**j**, **l** and *n*), Tukey's test for multiple comparison (*n*) and log-rank test (**g–i**, **k**, and **m**). Source data are provided as a Source Data file for **a**, **j–n** and **f**. Source data for **d–i** can be accessed at DOI 10.17605/OSF.IO/TNAX9.

readily manipulated, we repeated the in vitro experiments with a mixed population of MM cells that were either sensitive (U266; YFP positive) or resistant (U266-PSR; RFP positive). In cultures composed of 90% PI-sensitive U266, 10% PI-resistant U266-PSR, we observed that MSCs protect against BTZ, leading to enhanced relapse compared to MM cells cultured alone thus confirming our in silico observations (Fig. 6l, m). Next, we investigated the role of EMDR in vivo. Currently, there are no approved stromal targeting therapies available for the treatment of MM, however, Zol is frequently given to patients to inhibit osteoclast-mediated bone destruction that in turn also reduces the release of BDF that promote MM survival. Thus, NSG mice were inoculated with 90% PI-sensitive U266-GFP-Luc and 10% PI-resistant-RFP MM cells. Upon detection of the tumors in the hindlimbs, mice were divided in to two groups and pretreated with either vehicle or Zol for one week to limit osteoclast-mediated release of BDFs. Following pretreatment, mice were further subdivided and treated with BTZ only or BTZ and Zol for another week. As predicted by our *in-silico* results, limiting EMDR with Zol led to a deeper response to BTZ in PI-sensitive MM, as evidence by bioluminescence imaging and endpoint flow cytometry (Fig. 6n, o, supplementary fig. 9a–f). Whilst high concentrations (>10µM) of Zol have been shown to limit MM viability[62,63] in vitro, our results indicate that the combination of Zol (at clinically relevant concentrations[64]) and BTZ had no additional benefit to single agent BTZ in vitro (Supplementary Fig. 9g). However, in vivo we observed significant reduction in sensitive MM cell numbers in the combination group, suggesting that the anti-myeloma effect is not mediated through direct effects of Zol but more likely through altering of the local tumor microenvironment, which is in line with previous findings[65].

Our HCA model revealed during BTZ application, EMDR protected a reservoir of sensitive cells that ultimately develops intrinsic resistance leading to the increased relapse rates observed (Fig. 7a-b; Supplementary Fig. 10a). The role of this sensitive reservoir in relapse is particularly evident at the lower resistance probability, where tumors with EMDR exhibited more variable (generally longer) relapse times compared to tumors without EMDR. When EMDR is absent, many tumors go extinct and relapse only occurs if intrinsic resistance arises early following treatment, whereas with EMDR, sensitive cells persist, and resistance can arise later in the course of the disease. We validated this finding in vitro, where we demonstrate that MSCs protect a small portion of PI-sensitive U266 MM cells from BTZ treatment, whereas in cultures containing only MM cells, PSR-RFP MM cells ultimately became the sole clone upon relapse following BTZ treatment (Fig. 7c, d). Further, in vivo, treatment of mice with BTZ, led to the outgrowth of PI-resistant PSR-RFP. However, in Zol-pretreated (EMDR-inhibited) mice we see a deeper reduction in PI-sensitive MM cells compared to BTZ-only treated mice, where EMDR is intact (Fig. 7e, f). We next determined the clinical relevance of whether PI sensitive MM

cells existed in patients that were considered to have relapsed refractory MM (RRMM). We isolated CD138+ cells from MM and exposed the cells to BTZ ex vivo. We observed that approximately 20% of relapsed refractory MM patients who had failed a PI-containing regimen (*n* = 86) contained MM cells that are still sensitive to PI treatment thus supporting our in silico, in vitro and in vivo findings. Taken together, these results demonstrate the ability of the HCA to predict the effect of resistance and EMDR on tumor relapse and indicate a vital role of EMDR in contributing to minimal residual disease and the development of relapse refractory MM.

### Environment-mediated drug resistance and resistance probability impact MM heterogeneity in relapsed disease

Even though relapsed tumors with a high resistance probability had similar growth dynamics with and without EMDR, we noted that the composition of the tumors differed greatly. Myeloma is a clonal disease with the prevalence of clones changing throughout treatment[23,49,66,67]. Given the spatial resolution of the HCA model, we next asked whether EMDR impacted the heterogeneity of resistant tumors. To this end, we tracked the number and spatial localization of new clones forming from unique resistance events under BTZ treatment conditions in the presence or absence of EMDR. We observed that at a high resistance probability ($p_\Omega = 10^{-3}$) EMDR drove the evolution of significantly more resistant subclones (mean = 5.2 subclones with >10 cells, *n* = 24 simulations) compared to non-EMDR conditions where the arising population was composed largely of a single resistant clone (mean = 1.2 subclones with >10 cells, *n* = 10 simulations; Fig. 8a), with similar results at lower resistance probabilities and pulsed BTZ treatment (Supplementary Fig. 10b, c). In vivo, we observed a greater reduction in the ratio of GFP+ sensitive to RFP+ resistant MM when mice were treated with Zol+BTZ compared to BTZ only (Fig. 8b, c). Moreover, the HCA revealed a significant proportion of resistant subclones originated in close proximity to MSCs/pre-osteoblasts (Fig. 8d) or following release of BDF from osteoclastic bone resorption (Supplementary Fig. 10c). Taken together, these data would indicate that EMDR not only leads to enhanced relapse, but increased MM heterogeneity due to the emergence of independent resistant MM clones. Further, it is also plausible that EMDR may give rise to MM clones with potentially unique intrinsic BTZ resistance mechanisms making the treatment of relapsed tumors more challenging. Together, these data support that early targeting of bone marrow microenvironment, and thus EMDR, would lead to reduced MM heterogeneity and potentially deeper responses with subsequent lines of treatments.

## Discussion
Understanding the factors that govern the evolution of refractory MM remains of huge clinical importance. Dissecting how intrinsic and bone

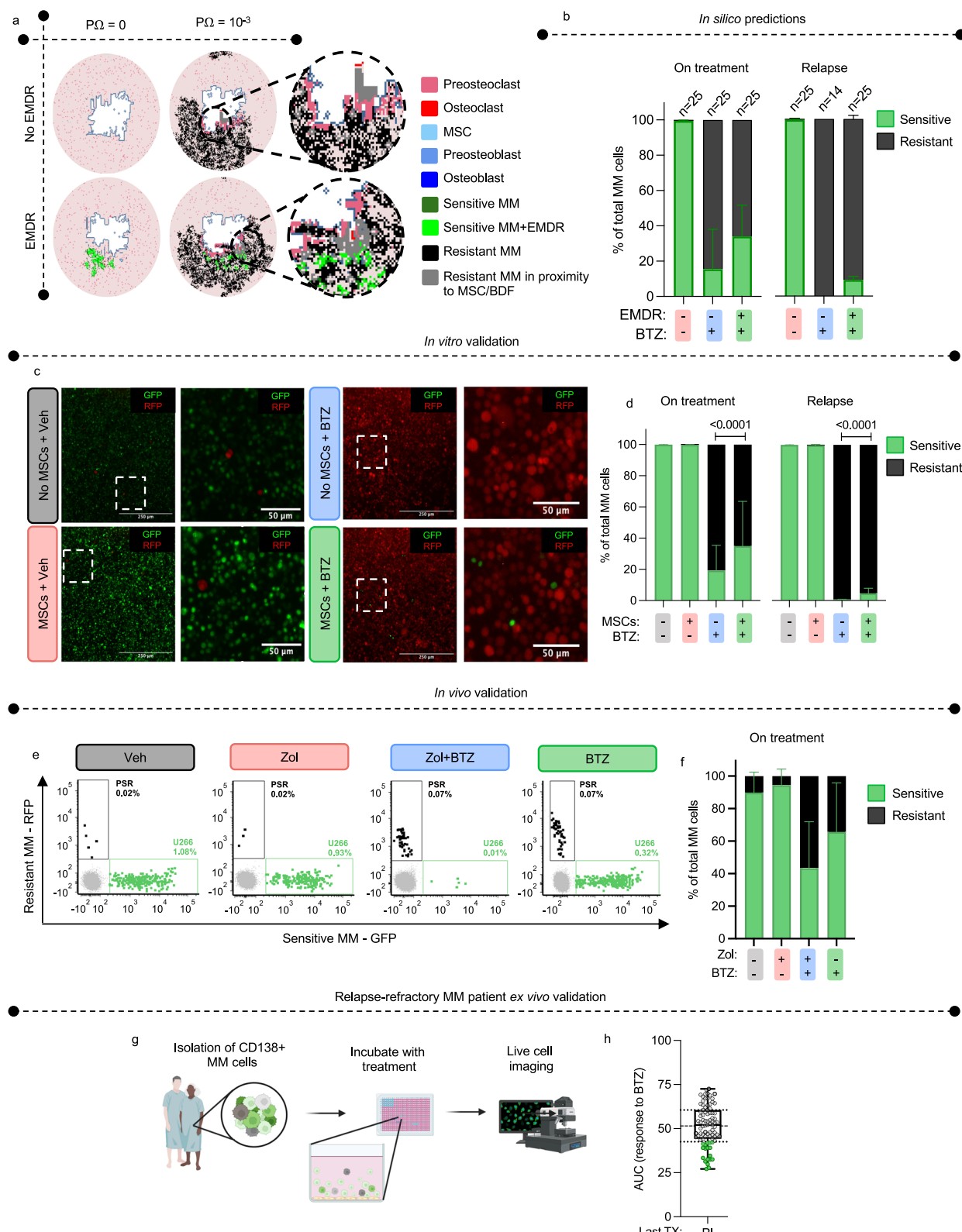

ecosystem mechanisms contribute to the process in the presence or absence of treatment or where these clones arise spatially are difficult questions to address with standard experimental techniques alone. Here, we address this using an experimentally derived hybrid cellular automaton (HCA) model of MM evolution in the bone ecosystem that provides insights into the development of relapsed-refractory disease. For example, we show that the bone ecosystem contributes to minimal residual disease, and MM clonal heterogeneity. These findings are in part supported by previous studies revealing enrichment of binding proteins and chemokine receptors on MM cells in minimal residual disease, indicating a likely role for stromal interactions in facilitating relapse[68]. Our findings and biological validation provide rationale for the targeting of the bone ecosystem during the application of standard of care treatment to prevent the emergence of multiple resistant MM clones.

**Fig. 7 | EMDR contributes protection of sensitive MM and tumor heterogeneity. a** HCA images from simulations with continuous BTZ application, with no resistance probability ($p_\Omega = 0$) or high resistance probability ($p_\Omega = 10^{-3}$) in the presence or absence of EMDR. Gray resistant myeloma cells are near MSCs or BDFs but not protected by EMDR. **b** Computational outputs of the proportion of BTZ-sensitive and BTZ-resistant MM cells 1 year on treatment (left) and at relapse (right) with $p_\Omega = 10^{-3}$. Values are mean ± SD. Related to Figs. 6f and i. **c** Representative images of U266-nuclear eYFP+ and PSR-RFP+ cells at relapse when cultured alone or huMSCs, with or without BTZ in vitro. Related to Fig. 6j, k. **d** The proportion of BTZ-sensitive (U266-nuclear eYFP+) and BTZ-resistant (PSR-RFP + MM cells cultured alone or with huMSCs with and without BTZ on treatment (left) and at relapse (right). Related to Figs. 6l, m and 7d. Values are mean ± SD. **e** Representative flow plots of BTZ-sensitive (U266 GFP+Luc+; green) and BTZ-resistant (PSR-RFP+; black) from the bone marrow (gray) of NSG mice treated with vehicle, Zol, Zol+BTZ or BTZ. Vehicle = 10 femur, Zol = 10 femur, Zol+BTZ = 12 femur, BTZ = 8 femur. **f** Proportion of BTZ-sensitive (U266 GFP+Luc+) and BTZ-resistant (PSR-RFP+) MM cells in the bone marrow of NSG mice treated with Vehicle = 10 femur, Zol = 10 femur, Zol+BTZ = 12 femur, BTZ = 8 femur. Values are mean ± SD. **g** Schematic of the EMMA platform. CD138 + MM cells and stroma are isolated from MM patients and co-cultured with test compounds. Live cell imaging is used to assess viability. Created with biorender.com. **h** The AUC of MM cells from RRMM patients ($n = 86$ patient samples), whose last relapse was to a PI-containing treatment (last TX) in response to BTZ treatment ex vivo. Data are presented as a box plot (centre line at the median, upper bound at 75th percentile, lower bound at 25th percentile with whiskers at minimum and maximum values). Each dot represents one MM sample. Green dots identify sensitive MM samples (quartile 1). White dots identify samples in quartiles 2–4. Statistical significance was determined by two-way ANOVA Šídák's multiple comparison (**d, f**). Source data are provided as a Source Data file for **d, f, h**. Source data for **a, b** can be accessed at DOI 10.17605/OSF.IO/TNAX9.

Our own in vitro and in vivo data, as well as data from the literature, were used where possible to parameterize and calibrate the HCA model. However, there were some parameters that could not be estimated, and the specific assumptions that have been incorporated into the HCA may impact the generality of the results. For example, we assume that MSCs are recruited in proportion to tumor burden based on experimental data in non-treatment conditions, but the application of treatment could alter this recruitment rate and thus bone ecosystem protective effects. Furthermore, there are limitations to how well parameters estimated from in vitro data correspond to the in vivo setting. To address this, we tested the robustness of the model results by varying a subset of parameter values to determine the impact on simulation results (Supplementary Methods 1.4; Supplementary Fig. 11). Overall, our finding that EMDR increases minimal residual disease and leads to higher relapse rates was consistent across parameter values, underscoring the robustness of the model.

There have been several mathematical models that explore interactions between MM and the bone marrow microenvironment. For example, ordinary differential equation (ODE), evolutionary game theory (EGT), and hybrid agent-based models have revealed insights into MM growth in bone and have even taken into consideration the impact of MM progression on bone destruction[69–75]. Additionally, mathematical models have focused on how ecosystem effects contribute to drug resistance[76–79]. However, the HCA model presented herein represents a significant advance in that it is carefully integrated with biological data to recapitulate the bone-MM ecosystem under normal and treatment conditions and is able to dissect intrinsic vs. ecosystem driven resistance in the disease. A major additional advantage of our model is its spatial nature, allowing us to visualize and measure where the ecosystem, and particularly stromal cells, contribute to the generation of resistant clones.

Our model shows that resistant MM clones can arise independently of the bone ecosystem during BTZ treatment if the resistance probability is sufficiently high. But, when EMDR is present, we observe the evolution of higher numbers of resistant clones near MSCs/pre-osteoblasts/BDF during continuous BTZ treatment. The model also suggests that targeting of EMDR during treatment would reduce MM heterogeneity. Various adhesion molecules and cytokines such as the CXCL12/CXCR4 and IL-6/JAK/STAT axis have been shown to mediate ecosystem protective effects and drugs/biologics that specifically target those pathways have been trialed[80,81]. Unfortunately, these drugs have had little success translating to clinical practice due to limited survival improvements when added to existing therapies[82], thus there remains a need to identify multiple mechanisms of EMDR in patients with feasible methods of delivering novel therapies that target them. However, we posit that early intervention with EMDR-targeting agents, perhaps in combination, would reduce tumor heterogeneity, rendering relapsing tumors vulnerable to a targeted second-line therapy that targets the predominantly homogenous disease. As such, future studies/trials should take into consideration the tumor diversity upon relapse and response to second-line therapies.

Another important aspect of the model is the ability to apply and withdraw therapy at varying stages of MM progression to quantitatively determine the impact on MM over time which may be useful for adaptive therapy design and implementation. Adaptive therapy is an emerging, evolution-inspired approach for cancer treatment that exploits competitive interactions between drug-sensitive and drug-resistant cells to maintain a stable tumor burden[61]. Several models have studied the many ways therapies can be applied in an adaptive manner ensuring that the competition between treatment-resistant and naïve populations allows for better overall survival for cancer patients. Importantly, our model captures tumor interactions with the bone ecosystem and thus includes the potential impact of the bone ecosystem on adaptive therapy. This allows us to study not only how treatments impact tumor populations but also their influence on bone ecosystem cells such as osteoblasts and osteoclasts, and how this can then influence MM response since the protective bone microenvironment can serve as a reservoir of sensitive cells under treatment. Clinical trials thus far using adaptive therapy guided by mathematical modeling have had encouraging results. For instance, recent studies have shown that in patients with bone metastatic prostate cancer, adaptive therapies guided by mathematical models can increase the time to progression and the overall survival using standard of care treatments[83]. These data support the feasibility of integrating mathematical modeling for the design of patient-specific treatment. It should be noted however, the HCA model is parameterized with a variety of in vitro, in vivo, and human data, making translation directly to the clinic challenging in its current form. This can be resolved with further rigorous model calibration and validation using clinical data, and multiple efforts at moving agent-based models into the clinical setting are underway[29–31,84,85].

A key component of our HCA model lies in its ability to recapitulate the homeostatic nature of bone remodeling over prolonged periods (~4 years). As we demonstrated, the impact of therapies on tumor-naïve bone volume and the bone stromal cellular components that control the process can also be quantitatively examined. It also can examine non-cancerous bone diseases that result from an imbalance in bone remodeling such as osteoporosis[86]. The underlying causes of osteoporosis can result in differing effects on osteoclasts and osteoblasts, which could be incorporated into the model. For example, estrogen deficiency increases the lifespan of osteoclasts while having the opposite effect on osteoblasts, whereas age-related senescence is characterized by decreased osteoclastogenesis and osteoblastogenesis[86]. Osteoporosis treatments include anti-resorptive therapies (e.g., bisphosphonates) and bone formation therapies (e.g., intermittent PTH). By modifying certain model parameters, we can explore the effect of different therapeutic strategies

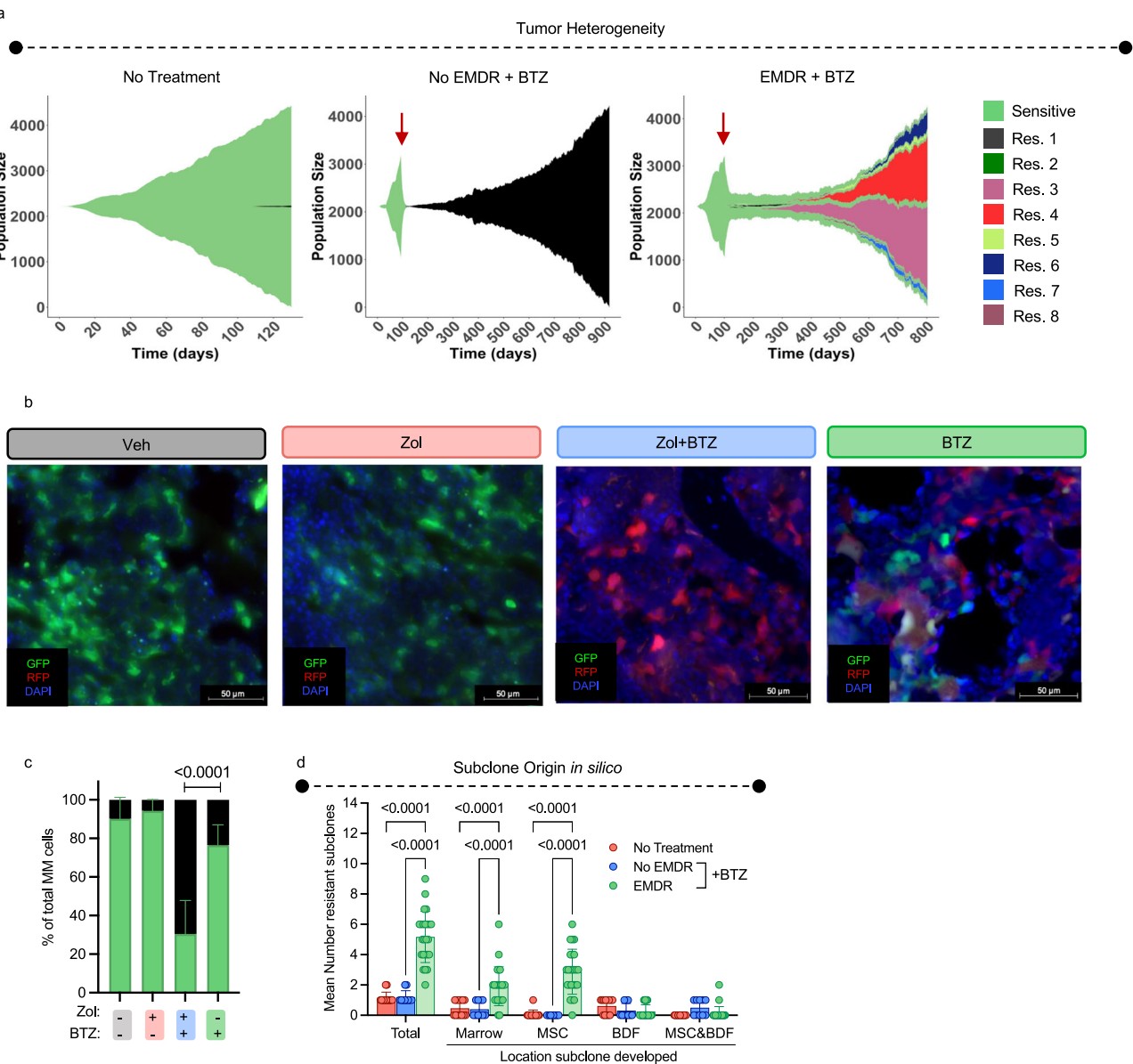

**Fig. 8 | EMDR contributes to tumor heterogeneity upon relapse. a** Muller plots of sensitive and individual BTZ resistant (Res) sub-clones from simulations described in a-c. Colors denote different subclones. Red arrow indicates start of BTZ treatment. **b** Images of GFP + U266 (green) and RFP + PSR (red) MM cells and cell nuclei (DAPI; blue) in tibial sections of mice from in vivo study described in Fig. 7. Magnification 20X. Scale bar 50 microns. **c** Relative quantification of GFP + U266 and PSR MM in tibial sections described in Fig. 8b. Vehicle = 5 tibia, Zol = 5 tibia, Zol+BTZ = 7 tibia, BTZ = 4 tibia. Values are mean ± SD. **d** Mean number of resistant sub-clones arising following BTZ treatment tumors that reached 20% with resistant subclones and the locations within the bone marrow microenvironment where each subclone originated, with/without EMDR from simulations described in a. The *n* number represents the number of simulations that developed resistance out of the 25 independent simulations. *n* = 13/25 (No Treatment), *n* = 10/25 (No EMDR + BTZ), *n* = 24/25 (EMDR + BTZ). Values are mean ± SD. Statistical significance was determined by two-way ANOVA with a Šídák's multiple comparison (**c**) or Tukey's test (**d**). Source data are provided as a Source Data file for **c** and **d**. Source data for **a** can be accessed at DOI 10.17605/OSF.IO/TNAX9.

on disease outcomes to determine if certain treatments are more effective in particular scenarios[87]. While we have primarily used Zol and BTZ for proof of principle, the model can be easily adapted to integrate the effects of other standard-of-care therapies as single agents or in combination such as dexamethasone, and IMIDs. Importantly, the temporal and cellular resolution of the model also can identify key moments in which to administer treatments to maintain a healthy bone volume.

Our HCA model, like all models, has caveats. For example, its two-dimensional nature does not take into account the potential three-dimensional role in evolutionary dynamics[88,89] or the vascular nature of the bone marrow-MM microenvironment that can play a role in

drug/nutrient diffusion[10,90], nor the adiposity, nor the extracellular matrix of the microenvironment, which can have a profound effect on MM cell growth. However, these can be incorporated into the model, if necessary, upon availability of relevant experimental data. Another potential and future extension of the model is to incorporate the immune ecosystem given its role in controlling bone turnover and cancer progression[10,90], the influence of BDFs on immune populations[91,92] and the growing armamentarium of immunotherapies in MM. Here, we excluded the immune component to focus solely on the role of the bone stroma on MM progression combined with the fact that MM growth in immunocompromised mice was used to parameterize and compare model outputs. Infiltrating immune cells

have noted effects on cancer progression. For example, initially, natural killer (NK) cells and cytotoxic T lymphocytes can drive an anti-tumor response[93]. However, as the tumor progresses, immunosuppressive populations expand including myeloid-derived suppressor cells (MDSCs) and regulatory T-cells (Tregs)[93]. MM is characterized by an increase of inactive NK cells, MDSCs, and Tregs and treatments such as bortezomib can alter the composition and activity of infiltrating or resident immune cells[93,94]. Tumor-immune dynamics in MM have been previously explored with ODE models[95,96] and parameters derived from in vitro or in vivo studies can be integrated into our HCA model albeit with a limit on complexity. Another caveat is the difficulty in determining the rate of MM resistance and the assumption that it occurs intrinsically due to treatment pressure. Prior studies suggest that drug resistance may arise through the selective expansion of pre-existent resistant tumor cells, de novo, or through the gradual increase in resistance over time[97]. In our model, we assume that after treatment initiates (i.e., once the bone marrow consists of 10% of MM cells), there is a probability that a dividing MM cell could develop a resistance mechanism, and that mechanisms directly causes resistance to bortezomib resulting in heterogeneity. This resistance probability that would lead to treatment resistance is difficult to assess in patients or in experimental models but the resultant heterogeneity arising from different resistance probabilities may allow us to infer resistance rates in patients. Moreover, the in silico model is not fixated on specific resistance mechanisms currently but more on whether a cell is sensitive or not to applied treatment. One such benefit of the in silico model is the ability to alter parameters, such as $p_\Omega$, response to treatment and doubling time, to mimic the heterogeneity observed clinically.

In conclusion, we have described a mathematical model of MM evolution in the context of the bone ecosystem. Our results show that, under treatment conditions, the interactions between the bone ecosystem and the tumor contributes to the presence of residual disease and to the enrichment of tumor heterogeneity. These results suggest that early intervention with drugs that target the bone ecosystem could lead to MM extinction or at the very least reduce tumor heterogeneity by preventing the evolution of multiple drug-resistant clones. Given the current interest in novel strategies of treatment that consider the evolutionary dynamics in cancer such as adaptative therapies, we also predict that our model, armed with future robust clinical data and validation could be a useful tool in guiding patient-specific treatment strategies.

## Methods

### Hybrid Cellular Automaton Model
The model we developed builds on the HCA paradigm originally described by Anderson et al. and used to study evolutionary dynamics in cancer and our work in the context of bone metastasis[40–42,98,99]. By definition, an HCA consists of discrete cell types that are updated once per time step according to a set of flow charts (Fig. 1c) as well as a set of partial differential equations that describe cytokines in the microenvironment. Here, we consider six different cell types, including precursor osteoclasts, active osteoclasts, mesenchymal stem cells (MSCs), precursor osteoblasts, active osteoblasts, and multiple myeloma cells. Additionally, we incorporate two signaling molecules including RANKL and bone derived factors (BDFs) such as transforming growth factor beta (TGF-β). In the sections below, we describe the interactions between cell types and cytokines that are key in bone remodeling and MM progression. Further details can be found in Supplementary Methods 1.

### Parameters and grid
We implemented our model using the Hybrid Automata Library (HAL)[100]. When possible, parameters for the HCA model were derived from empirical and published data (Supplementary

Tables 1–6). The model is defined on a 2D rectangular grid (160 × 150 pixels) representing a $1600 \times 1500 \mu m^2$ cross section of the bone marrow. Trabecular bone is defined in the center of the grid and initially consists of 12.9% of the total area, consistent with our in vivo data (Supplementary Methods 1 1). Normal bone remodeling events are uniformly distributed over 4 years, the approximate turnover time of trabecular bone[86]. A remodeling event is initiated by the expression of RANKL by five osteoblast lineage cells on the perimeter of the bone.

### Preosteoclasts and osteoclasts
Precursor osteoclasts follow the gradient of RANKL to the bone remodeling site. With a given probability, at least five preosteoclasts will fuse together to become a multinucleated osteoclast. Active osteoclasts resorb bone matrix, leading to the release of BDF. We assume that the amount of bone that osteoclasts resorb is proportional to their lifespan of approximately 14 days[86,101].

### MSCs, preosteoblasts, and osteoblasts
After an osteoclast fuses, an MSC is placed on the grid within a radius of 40 μm of the osteoclast, provided there is not already an MSC within the neighborhood. This requirement ensures that MSCs are located adjacent to sites of bone remodeling and can couple bone resorption with bone formation[36]. MSCs undergo asymmetrical division to create preosteoblasts, which proliferate when BDF is above a certain threshold or differentiate into bone matrix-producing osteoblasts when BDF is below the threshold (Supplementary Fig. 2c). We assume that osteoblast lifespan is proportional to the amount of bone that was resorbed at the location of the osteoblast, which is approximately 3 months. In the model, we assume that osteoblast death is proportional to the amount of bone that was resorbed by the osteoclast subunit that had occupied the space. Whilst this is a simplifying assumption of the biology, this assumption permits the coupling of bone resorption with bone formation, since the lifespan of an osteoblast controls how much bone the osteoblast forms. Bone resorption and bone formation are tightly coupled processes[102]. The definitive mechanisms through which osteoclasts and osteoblasts sense each other and their catabolic/anabolic processes have still not been fully elucidated and can be context dependent. However, several mechanisms are known to play a role including the release of bone derived factors (BDFs; namely TGF-beta and IGF-1), osteoclast derived factors such as sphingosine 1-phosphate, extracellular vesicles, apoptotic bodies containing RANK, and demineralized collagen remnants in the resorption lacunae. To model each of these mechanisms independently would make our model very complex as well as computationally expensive. We therefore integrated a straightforward coupling of osteoblasts and osteoclasts in the model. Further, when an osteoblast is buried, this has consequences for bone homeostasis. To maintain a steady state of bone, the lifespan of the nearest osteoblast is increased to allow it to build more bone to compensate for the amount lost due to the buried osteoblast. Whilst this is a simplifying assumption, research suggests osteoblast lineage cells communicate through gap junctions. This allows osteoblasts to coordinate activities and ensure new bone tissue is deposited at the correct location and in the appropriate amount to maintain structural integrity and bone homeostasis[103–107].

### Cytokines
RANKL ($R_L$) and BDF ($T_\beta$) are two key cytokines that drive the normal bone remodeling process. RANKL is produced by osteoblast lineage cells ($\alpha_R B_{i,j}$) with natural decay of the ligand ($\delta_R R_L$). BDF is released during bone resorption ($\alpha_T B_{i,j} C_{i,j}$) with natural decay of the ligand ($\delta_T T_\beta$). We also assume that there is a constant production of BDFs ($\alpha_B$) by other cell types that are not explicitly defined in the model. These cytokines are incorporated into our model through the

following system of partial differential equations (PDEs):

$$\frac{\partial R_L(x,y,t)}{\partial t} = \underbrace{D_R\left(\frac{\partial^2 R_L}{\partial x^2} + \frac{\partial^2 R_L}{\partial y^2}\right)}_{\text{Diffusion}} + \underbrace{\alpha_R B_{i,j}}_{\text{Production}} - \underbrace{\delta_R R_L}_{\text{Decay}}$$

$$\frac{\partial T_\beta(x,y,t)}{\partial t} = \underbrace{D_T\left(\frac{\partial^2 T_\beta}{\partial x^2} + \frac{\partial^2 T_\beta}{\partial y^2}\right)}_{\text{Diffusion}} + \underbrace{\alpha_B}_{\substack{\text{Basal}\\\text{Production}}} + \underbrace{\alpha_T B_{i,j} C_{i,j}}_{\text{Production}} - \underbrace{\delta_T T_\beta}_{\text{Decay}}$$

(1)

where the subscripts specify the location on the grid, i.e., $x = ih$, $y = jh$ where $\{i, j\}$ are positive integers and $h$ denotes the spatial step, $h = \Delta x = \Delta y$. The system of PDEs is solved using the forward time centered space (FTCS) scheme with periodic boundary conditions imposed on all sides of the domain, which is implemented using the diffusion function in HAL.

## Multiple Myeloma

A single myeloma cell is recruited within a radius of 40 $\mu m$ of an osteoclast after the first osteoclast fusion event occurs (Fig. 4a). MM cells move and proliferate in response to BDF and have a further proliferative advantage if they are within a radius of 20$\mu m$ of an MSC or preosteoblast (Fig. 3g), and a survival advantage if BDF is above a certain threshold (Fig. 3h). As the MM cell population increases, additional MSCs are recruited. Furthermore, if three or more myeloma cells are located close to bone, a new bone remodeling event is initiated which increases RANKL and osteoclast formation. MM cells also prevent the differentiation of preosteoblasts if they are located within a radius of 80$\mu m$ of a MM cell which decreases osteoblast activity.

## Treatment

The HCA incorporates the direct and indirect effects of bortezomib by including the cytotoxic effect on MM cells as well as its ability to inhibit preosteoblast fusion and enhance bone formation[47,48] (Supplementary Fig. 3). To mimic the clinical scenario of MM diagnosis, once MM reaches 10% of the bone marrow area in silico, treatment is applied either continuously (as for mouse models of multiple myeloma) or pulsed (2 weeks on, 1 week off; as is performed in the clinic) until MM burden reaches 20% of the bone marrow. In these simulations, once treatment is initiated, myeloma cells have a probability of developing resistance mechanisms during cell division that would cause the MM cells to become resistant to bortezomib. MM cells that are within a radius of 20 $\mu m$ of an MSC or preosteoblast or are located at a position where BDF is above a certain threshold, have a survival advantage during treatment referred to as environment mediated.

## Cell culture

Human proteasome inhibitor sensitive U266 myeloma cell and their proteasome inhibitor resistant derivatives (PSR[108,109]) were a kind gift from Dr. Steven Grant at the University of Virginia, VA. U266 and PSR were transduced using QIAGEN lentiviral particles (CLS-PCG-8 or CLS-PCR-8) according to manufacturer's instructions to express GFP and RFP respectively, generating U266-GFP and PSR-RFP. These cells were cultured in RPMI containing 10% FBS (PEAK), 1% penicillin-streptomycin. MCSF-generated macrophages were isolated from tibia and femur harvested from 6-week-old C57Bl/6 RAG2$^{-/-}$ mice[110,111]. Bone marrow cells were collected by centrifugation (10,000 g, 15 s). Isolated cells were plated in αMEM (+/+) containing 10% FBS (Peak), 1% penicillin-streptomycin and MCSF (300-25, Peprotech; 30 ng/ml). Adherent macrophages were collected and used for downstream experiments after 72 hours. Murine mesenchymal stromal cells (MSCs) were isolated tumor naive 4–6-week-old male and female C57/BL6 Rag2$^{-/-}$ mice[112]. -. Following removal of muscle tissue from the long bones, epiphyses were removed and bone marrow was depleted by centrifugation at 10,000 $g$ for 15 seconds. Flushed bones were then cut into 1–3 mm bone chips. The bone fragments were digested for 1 hour at 150 rpm, 37 °C in 1 mg/mL collagenase II (Invitrogen) in α-MEM with 15% FBS. The digested bone fragments were moved to 6-well tissue plates in 15% α-MEM, where the MSCs were allowed to migrate out of the bone chips and proliferate.

Human MSCs (PT-2501) and the murine preosteoblast cell line, MC3T3-E1 (CRL-2593) were purchased from Lonza and ATCC, respectively. Mouse MSCs and MC3T3-E1 cells were cultured in αMEM (-/-) containing 1% penicillin-streptomycin and 15 or 10% FBS (PEAK), respectively. Human MSCs were cultured as above except with 10% qualified FBS (Gibco). drug resistance (EMDR).

## Multiple myeloma mouse models

All animal experiments were performed with University of South Florida (Tampa, FL) Institutional Animal Care and Use Committee approval (CCL; #7356R, #10955R). All mice were kept in a 12 h light/dark cycle in ambient humidity and temperature. Male and female 14–16-week-old immunodeficient mice NOD-SCIDγ (NSG; Jackson Laboratories, Strain #:005557, RRID:IMSR_JAX:005557) mice were divided into tumor naïve or tumor bearing mice GFP-expressing human U266 (U266-GFP) multiple myeloma cells were injected (5 × 10$^6$cells/100 μL PBS) or PBS (100 μL) via tail vein. Mice were euthanized at days 3, 10, 20, 40 and 100 (3-5/group/timepoint). Since MM is systemic, maximum tumor size is not used as an endpoint, rather hindlimb paralysis and >20% weight loss. Endpoints were only reached for 1 mouse in the day 100 group. In a separate experiment male and female 8-week-old NSG mice (Jackson Laboratories, Strain #:005557, RRID:IMSR_JAX:005557) were inoculated with PI-sensitive U266-GFP-Luc (90%) and PI-resistant PSR-RFP (10%) MM cells via tail vein injection (5 × 10$^6$cells/100 μL PBS). Upon detection of tumor burden in the hind limbs via bioluminescent imaging (IVIS), mice were randomized to two groups to receive either twice weekly vehicle (PBS) or Zol (30 μg/kg). Following one week of Zol pretreatment, mice were further randomized in to four subgroups and received the following: Group 1 Vehicle (5 mice), Group 2 Zol (30 μg/kg/twice weekly; 5 mice), Group 3 Zol+BTZ (7 mice), Group 4 BTZ (0.5 mg/kg/twice weekly; 4 mice). After one week of BTZ treatment, all mice were euthanized. Tibiae were excised and soft tissue removed for histological and FACs analyses. Since MM affects both sexes, disaggregated data is not provided.

## Micro-computed Tomography, Immunofluorescence and Histomorphometry

Harvested right tibiae were fixed in 4% paraformaldehyde (PFA) in 1x phosphate buffered saline (PBS) for 48 hours at room temperature. Evaluation of trabecular bone microarchitecture was performed in a region of 1000 μm, beginning 500μm from the growth plate using the SCANCO μ35CT scanner. Tibiae were decalcified in an excess of 10% EDTA at pH 7.4 at 4 °C for up to three weeks with changes every 3 days before being place in cryoprotection buffer (30% sucrose), frozen in cryomedium (OCT) and subjected to cryosectioning to yield 20 μm tissue sections. Sections were dried, washed and blocked before addition of primary antibodies (anti-pHH3; 1:200, 06-570 Millipore Lot # 2972863, anti-Osterix; 1:500, ab22552, abcam lot# GR3263824-6: anti-αSMA; 1:200, PA5-16697, Invitrogen). Slides were washed and incubated with goat anti-rabbit Alexa Fluor-647-conjugated secondary antibody (Invitrogen (#A21244) lot# 2277746 (1:1000)) and counterstained with DAPI. Mounted sections were imaged using 10X image tile-scans of whole tibia. pHH3+, αSMA+, Osterix+ or GFP+ cells were identified and quantified using Fiji software version 1.0[113]. Dual ALP and TRAcP staining were performed on cryosections. Brightfield visualization was performed using the EVOS auto at 20X magnification. Five images per section were taken and the number of ALP+ cuboidal bone-lining osteoblasts and red multinucleated TRAcP+ osteoclasts per bone surface were calculated.

**Reporting summary**

Further information on research design is available in the Nature Portfolio Reporting Summary linked to this article.

## Data availability

Biological data for all figures in this paper are provided in the accompanying Source data file. The raw images generated in this study are available freely by contacting either corresponding author Conor Lynch (conor.lynch@moffitt.org) or David Basanta (david@cancer-evo.org). No publicly available or previously published datasets were used in this study. The in silico data generated in this study have been deposited in the Open Science Framework (OSF) database under identifier DOI 10.17605/OSF.IO/TNAX9[114] [https://osf.io/tnax9/?view_only=7ff837c0a36e4b728791ec70f9935cd4]. The bone ecosystem facilitates multiple myeloma relapse and the evolution of heterogeneous drug resistant disease. *Open Science Framework*. DOI 10.17605/OSF.IO/TNAX9 (2023). Source data are provided with this paper.

## Code availability

The code generated in this manuscript is available at https://github.com/dbasanta/MM_ABM and has been deposited in the Open Science Framework (OSF) database under identifier https://doi.org/10.17605/OSF.IO/TNAX9[114] [https://osf.io/tnax9/?view_only=7ff837c0a36e4b728791ec70f9935cd4].

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

## Acknowledgements

The authors would like to thank the patients at H. Lee Moffitt Cancer Center and Pentecost Myeloma Research Center (PMRC). This work has been supported in part by the Small Animal Imaging Laboratory (SAIL), the Flow Cytometry core, the Cancer pharmacokinetics and pharmacodynamics core (CPPC), the Collaborative Data Services Core (CDSC) the non-therapeutic research office (NTRO), and the Analytical Microscopy Core (AMC) at the Moffitt Cancer Center, an NCI designated Comprehensive Cancer Center (P30-CA076292). This work was supported in part by funds from NCI-U01CA244101 (to C.C.L and D.B) and NCI-U54CA193489 (to K.H.S and A.S), NCI-R01CA239214 (to C.C.L), H. Lee Moffitt Cancer Center's Team Science Grant (A.S., K.H.S.), Miles for Moffitt Foundation (A.S.). Additionally, this work was supported by philanthropic support from the Pentecost Myeloma Research Center (to K.H.S., A.S. and others).

## Author contributions

A.K.M, M.F. and D.B. were responsible for computational model design and execution. R.T.B and C.C.L were responsible for biological experimentation design and analysis. T.L, N.N, K.N, M.N and J.F performed biological experiments. P.R.S, R.C, A.S and K.H.S were involved in the conception, design, experimentation, and analysis of ex vivo samples. A.K.M, R.T.B, C.C.L and D.B were involved in the overall concept and design. All authors were involved in manuscript writing and editing.

## Competing interests

K.H.S. reports honoraria from Bristol Myers Squibb, Janssen, Adaptive Biotechnology, Sanofi, GlaxoSmithKline, Takeda, Amgen, and Sebia as well as research funding to the institution from AbbVie and Karyopharm. All the other authors have no competing interests.

## Ethical Approval and Informed Consent

All animal studies were approved by the University of South Florida IACUC protocol #7356R and #10955R. Patient clinical samples for ex vivo assays and the corresponding clinical data was accessed through the Moffitt Total Cancer Care database. The Moffitt Total Cancer Care research protocol collects samples and biospecimens from cancer patients including multiple myeloma patients. Investigators obtained signed informed consent from all patients who were enrolled in Total Cancer Care® (protocols MCC14745 and MCC18608) at H. Lee Moffitt Cancer Center and Research Institute, as approved by the Institutional Review Board. Patient samples were used in accordance with the Declaration of Helsinki, International Ethical Guidelines for Biomedical Research Involving Human Subjects (CIOMS), Belmont Report, and U.S.

Common Rule. The medical records were deidentified and provided to us by an honest broker (Raghunandan Reddy Alugubelli) at the Collaborative Data Services Core at Moffitt Cancer Center. Patient derived myeloma cells ($n = 86$) were isolated from 68 patients, with a median age of 65.1 years (range, 36.0–82.8 years) with 51.16% of samples being from males and 48.84% from females. Neither sex, gender nor age analyses were taken into account. Only the following clinically relevant individual patient information was reviewed: the treatment administered (therapeutic agents, doses, and schedule) prior to biopsy.
