## [Peer Review File · Nature Communications]

REVIEWER COMMENTS

Reviewer #1 (Remarks to the Author): Expert in mathematical modelling and blood cancers

Multiple myeloma is an osteolytic plasma cell malignancy. The authors propose a hybrid cellular automaton model (agent based model coupled with reaction diffusion equations) describing the interaction of multiple myeloma cells with cells responsible for bone formation and degradation (active & precursor osteoblasts, active & precursor osteoclasts, mesenchymal stem cells) and prototypical signaling/growth factors (RANKL, TGF-beta). The authors show simulation results supporting that the parameterized model is able to recapitulate bone homeostasis. Based on the proposed 2d spatio-temporal model the authors simulate the impact of environment mediated drug resistance (EMDR) and mutation-induced drug resistance on minimal residual disease, treatment failure and myeloma cell heterogeneity.

The proposed manuscript contributes to hypothesis generation, however does not provide definitive insights into the mechanisms of resistance and relapse. I acknowledge that the literature research, the implementation of the model and its parametrization is a merit on its own right. Nevertheless, key findings presented in this manuscript are straightforward consequences of the model design and the model assumptions. As such the new biological insights are limited. A careful experimental or clinical validation of model predictions is not undertaken. Key results are qualitative without direct implications on the development of novel therapeutic concepts. Taking into account the high complexity of the model, the qualitative nature and the simplicity of the obtained results limit my enthusiasm for the proposed work.

I suggest to further elaborate on the following aspects:

(A) Important model assumptions are not well motivated, some model assumptions seem artificial and do, in my opinion, not reflect plausible biological processes.

- "We assume that each aOC resorbs bone in the direction with the maximum amount of bone." How can cells sense the direction with the maximum amount of bone? What is the biological mechanism or experimental observation underlying this assumption?

- "In the model, we assume that aOB death is proportional to the amount of bone that was resorbed by the aOC subunit that had occupied the space prior to the aOB (aOC_depth). Therefore, aOB death is also proportional to the average time it takes for an aOB to form one unit of bone (basal_time). This assumption permits the coupling of bone resorption with bone formation, since the lifespan of an aOB controls how much bone the aOB forms. Thus, bone homeostasis is maintained even when the amount of bone resorbed by an aOC varies."

This assumption seems rather artificial to me. How can aOBs sense how much bone aOCs have resorbed? I have the feeling that this ad hoc assumption is put forward to achieve realistic model behavior without having to care about more complicated regulatory mechanisms. However, this makes it very difficult to appreciate and interpret the properties of the proposed model. This assumption should be discussed in more detail and should be mentioned in the main text.

- "When an aOB is buried, this has possible consequences for bone homeostasis since the aOB did not form as much bone as it would have had it completed its full lifespan. To maintain a steady state of bone, the lifespan of the nearest aOB is increased to allow it to build more bone to compensate for the amount lost due to the buried aOB. This is a simplifying assumption of the model since biologically there are likely to be many factors involved to regulate the number and lifespan of osteoblasts."

This assumption seems rather artificial to me. How can aOBs know that an adjacent OB was buried? I am unaware of biological mechanisms that could explain such dynamics. Even if this assumption is required to obtain reasonable model dynamics, it weakens the mechanistic insights the model can provide. The necessity of such ad hoc assumptions might suggest that important biological mechanisms have not been sufficiently resolved.

- "After an aOC fuses, an MSC is placed on the grid within a certain radius of the aOC (MSC_radius), provided there is not already an MSC within the neighborhood. This requirement ensures that MSCs are located adjacent to sites of bone remodeling to couple bone resorption with bone formation 15 . However, to preserve that approximately 0.01% of the bone marrow consists of MSCs 2 , MSCs that are not within a certain radius of an aOC are removed from the grid. Therefore, the number of MSCs fluctuates over time but is capable of returning to the initial condition." This assumption seems rather artificial to me. Why is the spatial distribution of MSC and their co-localization with aOC not modeled based on first principles?

- The used mutation rates have to be motivated in more detail. How do the authors know at which rate mutations conferring BTZ resistance occur? How do they know that the considered orders of magnitude are biologically feasible?

(B) In my opinion, many model features and key results are straightforward conclusions of model assumptions (the authors get what they put in). Key dynamic features are not an emerging property which is based on more complex regulatory mechanisms but they are more or less directly prescribed. In this sense the first three sections of the results part provide only very limited biological insights. Their main merit (and from the modeling perspective this is a merit!) is to show that the given sets of

assumptions and model parameters lead to a plausible model behavior. Also the observations described in the other sections of the results part follow more or less directly from prescribed model features.

- "Our results show that without EMDR, BTZ completely eradicated the disease at high doses (Fig. 6D, Supplementary Fig. 3j), consistent with our in vitro findings. However, when EMDR effects are included in the model, BTZ reduced MM burden significantly, but the disease persisted at a stable, albeit lower, level and did not grow over time compared to existing in vivo data (Fig. 6C-d)." In my opinion, this finding is a direct consequence of the assumptions and the way the authors model EMDR (Supplement: "EMDR is when myeloma cells are transiently protected from Bortezomib due to factors or interactions with other cells in the bone microenvironment, which we define in our model to be when bone derived factors are above a certain threshold or MSCs or pOBs are within a certain radius of a myeloma cell 21").

In short, the authors assume that a certain percentage of MM cells does not respond to treatment (EMDR). Then they present simulations showing that under this assumption a certain fraction of the cells persists during treatment. Therefore, this observation does not yield new mechanistic insights, it just demonstrates that in a model where it is assumed that not all cells respond to BTZ, simulations show that not all cells respond to BTZ.

- If there is a mechanism (e.g. EMDR) which increases the number of cells surviving treatment, in my opinion, it is not surprising that (1) the MRD is higher, as argued above, and (2) relapses originate from multiple clones: the more cells survive, the more cells are at risk for acquiring mutations, consequently the more different clones with resistance mutations will be detected in average.

- "In the model, the recruitment of MSCs in silico increases over time (Fig 5a) as we and others have previously reported" This is, I think, the direct consequence of the assumption that MSC increase proportionally to MM cells (Supplement, p. 9 "We assume that one MSC is recruited per 50 myeloma cells, and that once it is recruited, an MSC remains in the bone marrow"). As such this results is a double check of model implementation but not more. It basically shows that the simulation evolves according to the underlying assumptions. A similar reasoning may apply to findings 2 and 3 on p. 8 of the main text.

- "Similarly, lowering TGF- β levels in silico led to a dramatic increase in bone volume (Supplementary Fig. 2b-e). Interestingly, the HCA results showed that when TGF- β levels dropped below 90%, there was significant bone loss (Supplementary Fig. 2b-e) due to abrogation of MSC proliferation leading to reduced numbers of preosteoblasts that could contribute to bone formation (Supplementary Fig.

2e). These data are reflective of the known biphasic roles of TGF- β on cell behaviors and processes at high and low concentrations", "As expected, BDF has a biphasic effect on bone in which low BDF results in bone loss due to lack of MSC/pOB proliferation and medium BDF results in bone growth due to increased mineralization"

These findings are, in my opinion, rather direct consequences of model assumptions and not properties emerging from first principles or from complex regulatory loops. The assumptions are:

++ Supplement p. 5 "We assume that high levels of BDF promote proliferation of pOBs whereas low levels promote differentiation 16"

++ Supplement p. 5 "We assume that MSCs divide only when BDF is above a certain threshold"

++ Supplement pp. 6-7 "This data shows how the mineralization rate compares to the control, specifically that it increases as TGF- β decreases (Supplementary Fig.

2a). "

In short, the authors ASSUME that decreasing TGF-beta levels increase bone formation and that very low levels deplete the cell types responsible for bone formation. Then the authors OBSERVE in their simulations low TGF-beta levels lead to increased bone formation and that very low TGF-beta values lead to practically no bone formation.

- "showing that ZOL targets actively resorbing osteoclasts as demonstrated by a reduction in bone resorption per OC (Supplementary Fig. 4h) while BTZ limits the fusion of OC precursors as shown by a reduction in cumulative osteoclast numbers with treatment (Supplementary Fig. 3g)." Also this is, in my opinion, a direct consequence of the assumptions.

- "We then repeated our in-silico experiments and observed that with the lower mutation probability ($p_{mut} = 10^{-4}$), EMDR increased the likelihood of tumor progression with more simulations likely to relapse (54.4%, $n = 125$) compared to tumors without EMDR (6.4%, $n = 125$; Fig. 6e and g; Supplementary Video 3). Similarly, with a higher mutation probability ($p_{mut} = 10^{-3}$) tumors were more likely to relapse when EMDR is present (100%, $n = 25$) compared to tumors without EMDR (56%, $n = 25$, Fig. 6f and h;

Supplementary Video 4)." In my opinion, it is not surprising that an increase of mutations conferring drug resistance increases the risk of relapsing disease. It is furthermore, I think, not surprising that relapse becomes more frequent if the number of cells protected from treatment (EMDR) and at risk of acquiring resistance mutations increases.

- "We observed that at a high mutation probability ($p_{mut} = 10^{-3}$) EMDR drove the evolution of significantly more resistant subclones (mean = 5.2 subclones with >10 cells, $n = 24$ simulations) compared to non-EMDR conditions where the arising population was composed largely of a single resistant clone (mean = 1.2 subclones with >10 cells, $n = 10$ simulations; Fig. 7c-d), with similar results at lower mutation probabilities and pulsed BTZ treatment (Supplementary Fig. 9b-c). Moreover, the HCA revealed a significant proportion of resistant subclones originated in close proximity to MSCs/pOBs (Fig. 7c) or following release of BDF from osteoclastic bone resorption (Supplementary Fig. 9c). Taken together, these data would indicate that EMDR not only leads to enhanced relapse, but increased MM heterogeneity due to the emergence of independent resistant MM clones." As described above this is, in my opinion, a direct consequence of the model assumptions.

- "Our data demonstrates that resistant disease cannot develop without MM intrinsic mechanisms." I think this conclusion is straightforward: If treatment resistance is restricted to limited niches such as in the concept of EMDR, there can be no unbounded expansion of cells in presence of treatment unless cells acquire additional (EMDR-independent) resistance mechanisms.

(C) Results have to be formulated in a much more decent and careful way.

- "For example, we show that the bone ecosystem contributes to minimal residual disease, and MM clonal heterogeneity" This is in my opinion not fully true. The authors propose a hypothetical mechanism by which the micro-environment protects MM cells. Simulation show that such a mechanism (straightforwardly) leads to more MRD and MM clonal heterogeneity. However, the authors provide no compelling evidence for the existence of such a mechanism in vivo. E.g., they do not show the existence of BTZ sensitive cells in BTZ refractory patients.

- "The similarities in trends between the in vivo model and the HCA outputs, indicated that the HCA model was accurately recapitulating the MM-bone vicious cycle." The authors come to the conclusion that the HCA model accurately recapitulates the MM-bone vicious cycle. However, there exist substantial differences between the model dynamics and experimental results shown in Fig 4: For most of the time the increase of MM/MA (panel b) shows an approximate linear increase in the simulations, whereas in the experiments (e) the increase looks clearly exponential. The same applies to the decay of BA/TA shown in panels (d) and (e). The dynamics between 0 and 30 days also look quite different in panels (c) and (f). I have the feeling that the authors may miss some mechanisms governing the dynamics of the MM cell expansion.

- Also in Figure 5 (pairs d,f and b,h) the simulations seem to differ qualitatively from the experimental data

- "When EMDR is absent, many tumors go extinct and relapse only occurs if

intrinsic resistance arises early following treatment, whereas with EMDR, sensitive cells

persist and resistance can arise later in the course of the disease. Taken together, these

results demonstrate the ability of the HCA to predict the effect of mutation probability and

EMDR on tumor relapse and indicate a vital role of EMDR in contributing to minimal

residual disease and the development of relapse refractory MM." This is a hypothetical finding which has not been validated experimentally.

(D) Key conclusions are qualitative, some of them not new. The authors do not use the full potential of their model.

- The authors state „A major additional advantage of our model is its spatial nature, allowing

us to visualize and measure where the ecosystem, and particularly stromal cells,

contribute to the generation of resistant clones.“ However they never compare the 2d simulations to 2d data. Why?

- the authors use population-based quantities to compare model simulation to experiments (Fig 4 & 5), although the model and the histology are 2d. To infer population based quantities the spatial model may be over-complicated and bearing the risk of over-fitting.

-"Together, these

data support that early targeting of bone marrow microenvironment, and thus EMDR,

would lead to reduced MM heterogeneity and potentially deeper responses with

subsequent lines of treatments." "In

conclusion, our data demonstrates a significant role for the bone ecosystem in MM

survival and resistance, and suggests that early intervention with bone ecosystem

targeting therapies may prevent the emergence of heterogeneous drug resistant MM."

This conclusion is not new. See. e.g., <https://www.nature.com/articles/s41467-020-19932-1>

Kawano Y, Moschetta M, Manier S, Glavey S, Görgün GT, Roccaro AM, Anderson KC, Ghobrial IM.

Targeting the bone marrow microenvironment in multiple myeloma. *Immunol Rev.* 2015 Jan;263(1):160-72. doi: 10.1111/imr.12233. PMID: 25510276.

From the side of drug-development the challenge of MM therapy is not a shortage of concepts. The main complication is to identify the proper compounds and a feasible way to deliver them (see. e.g.,

Federico C, Alhallak K, Sun J, Duncan K, Azab F, Sudlow GP, de la Puente P, Muz B, Kapoor V, Zhang L, Yuan F, Markovic M, Kotsybar J, Wasden K, Guenther N, Gurley S, King J, Kohnen D, Salama NN, Thotala D, Hallahan DE, Vij R, DiPersio JF, Achilefu S, Azab AK. Tumor microenvironment-targeted nanoparticles loaded with bortezomib and ROCK inhibitor improve efficacy in multiple myeloma. *Nat Commun.* 2020 Nov 27;11(1):6037. doi: 10.1038/s41467-020-19932-1. PMID: 33247158; PMCID: PMC7699624.) These issues are, unfortunately, not at all addressed

(E) The sub-model of bone remodeling has to be delineated from previous models by the same authors. What is actually new?

(F) I have the feeling that the authors did not invest sufficient efforts to validate the predictions of their model experimentally

- According to the model predictions (e.g., Fig 7b) BTZ sensitive MM cells should be detectable in BTZ refractory individuals. Why didn't the authors check this?

- the authors did not undertake gene sequencing (e.g., from patients at different disease stages) to validate their predictions of clonal heterogeneity.

(G) Some of the experimental setups require a more detailed discussion:

- "Based on our in vitro observations, we also incorporated a cost of resistance 46 in the model such that BTZ-resistant cells proliferate at a slower rate and are out-competed by PI-sensitive cells in the absence of treatment"

Can the authors be sure that the resistance mutations themselves leads to a competitive disadvantage? In my opinion it cannot be ruled out whether other differences (other mutations, epigenetic differences, metabolic differences) of U266 and PSR cells are responsible for the observed differences in proliferative fitness. Can the in vitro observations be reproduced in competitive transplantation assays? Can the fitness costs be observed also in case of cells harboring only the specific mutations required for resistance (e.g., mutations at drug binding site) but no additional mutations/aberrations? This should be tested e.g., in a CRISPR-based approach.

- The authors conduct experiments with TGF-beta. Can they really assume that the observed changes are representative for all bone derived factors?

Minor:

Suppl. Tables:

- Please make sure to describe how each parameter indicated as "Estimated" was estimated. I miss (or overlook) e.g., the explanation for **maxRANKL** or T_s

- It would be nice to see how many data points were used for the fits mentioned in the supplement and main text.

- Please specify the meaning of the black / red dots in Sfig 2 b c g

Fig 1b:

- please add a reference to each arrow and add in which experimental system the respective results were obtained.

- What are the units in equations 6-8?

- Equation 6 requires a more concise explanation

Reviewer #2 (Remarks to the Author): Expert in mathematical modelling and agent-based modelling

Key results:

Multiple myeloma (MM) is a treatable plasma cell cancer. However, people who suffer from this disease will eventually develop resistance to the various lines of therapy. In this article, the authors investigate the impact of the bone marrow microenvironment on treatment resistance of MM by constructing a mathematical model, a hybrid cellular automaton (HCA), parametrized to in vitro, in vivo, and human data including their own MM mouse model and cell cultures. Interestingly, their model can capture the positive feedback on MM growth caused by the bone marrow environment. Their results suggest that environment mediated drug resistance (EMDR) protects a small number of MM cells from treatment, contributing to residual disease. Furthermore, they found that EMDR promotes heterogeneity in treatment-resistant MM cells, increasing the challenge of implementing efficient relapse therapies.

Validity:

For the most part, the manuscript is quite robust. I understand that the objective of the mathematical model presented is to capture key outputs and not to precisely fit the experimental results. However, I enquire about the criteria to accept the model's results as a valid representation of the phenomenon as some of the simulations are not measured in the same units or span the same time scale. Furthermore, from Fig 3, I wonder if it was tested that the protective effect is entirely due to bone derived factors (BDFs). What happens when those BDFs are inhibited? Will the spatial distribution of the apoptotic and dividing cells change? In regards the fact that the immune component is omitted in the study, one of the representative BDFs is TGF- β , a cytokine with many roles in immune regulation. What can be said about the effects of TGF- β on the immunodeficient mice model used?

Significance:

The results show that even with the immune system excluded from the study, the bone marrow environment is a non-negligible aspect of treatment resistance in multiple myeloma. Therefore, it further supports the idea of targeting the tumour microenvironment in second-line therapies. They also give us insights on the mechanisms leading to the development of heterogeneity and resistance in multiple myeloma.

Data and methodology:

In the methods, what is the reason for choosing the two types of treatment administration used for the HCA model? For the results, in Fig. 7a, I am confused as to why the simulation shows that there is

resistant MM with EMDR (grey, resistant MM + EMDR) in the case where there is an absence of EMDR (no EMDR). There are also small details I noticed about the presentation of the figures. In Fig. 1b, the black arrows are not defined. Fig. 4 b-g, Fig. 5 a-d, and Fig. 7b and 7d have missing or incomplete legends. I was not provided the coding portion of the computational algorithm, so I cannot comment on it.

Analytical approach:

Like mentioned above, there is no mention of the method used to compare the results of the HCA model to the experimental data. It appears to be a fully qualitative comparison, and maybe a quantitative test could improve the strength of the analysis.

Suggested improvements:

I have no suggestion for additional experiments or simulations.

Clarity and context:

The overall clarity and accessibility of the text are quite good, but there are some things to note.

1. A few acronyms should be defined before their use (BCMA, RANKL, CAR, TGF- β) and some can be dropped altogether, such as OC for osteoclasts, to help with the flow in the reading.
2. In the research questions found in the introduction, it asks how EMDR and intrinsic drug resistance both contribute to the heterogeneity of the disease in control and treatment conditions. Explicitly, what is the control condition in this situation? Furthermore, it says that the three research questions are difficult to address using current *in vivo* and *in vitro* techniques. It would feel more complete to explain and show cited examples as to why.
3. As the journal is not specifically targeted to people familiar with mathematical models, I think a separate sentence for the description of hybrid cellular automata models, instead of using parentheses, would give more context in the introduction.
4. In the HCA model, there is significant bone marrow loss when TGF- β levels are dropped by 90%. This is in comparison to normal physiological values?
5. In the discussion, there is a description of different mathematical models that explore the interaction between MM and its microenvironment for which the objective is not clear. It could be rewritten to explain, for example, if it is to demonstrate the novelty or the importance of the HCA model.

References:

The manuscript referred to the literature appropriately.

Your expertise:

My expertise is in computational medicine and mathematical modelling; therefore the *in vitro* and *in vivo* methods are outside the scope of it.

Reviewer #3 (Remarks to the Author): Expert in multiple myeloma genomics, therapy, and pathogenesis

This is an elegant model of the interaction between myeloma tumor cells and the bone microenvironment, incorporating in vitro and in vivo observations into in silico modeling. It is clinically relevant, particularly given that the model addresses potential mechanisms supporting minimal residual disease persistence.

My comments are minor:

1. I think it is an overstatement to say that MM resistance can't develop without cell intrinsic mechanisms. Your model includes mutations as only mechanism of resistance available- it is not including the role of the immune microenvironment nor any non-mutational mechanisms described to be associated with PI- resistance ie changes in signaling pathways / gene expression / metabolism. I would change this phrasing in the abstract
2. For all in vivo studies please include the number of mice in the figure legends.
3. In Figure 4 it appears that the bone destruction occurs ahead of the tumor expansion. How do you explain this? And how can you say it's myeloma-induced bone loss if the loss is mainly prior to 10% GFP+ cells?
4. In Figure 6, you model the effect of bortezomib starting at 10% PC only- how does the model behave if you have a higher % of MM prior to treatment?
5. Bortezomib-resistant cells do not always proliferate more slowly- what happens in the model if you remove the cost of resistance?
6. You state that early intervention with bone ecosystem targeting therapies may prevent the emergence of heterogenous drug resistant MM- can this be tested in silico in your model?
7. Are you able to show data with dexamethasone? Clinically relevant given that it's in most MM regimens, and biologically relevant given that it affects both MM and bone architecture
8. The discussion has a good description of the model limitations, I think it's well written

Author's Rebuttal

We thank the reviewers for their incredibly valuable feedback and excellent comments. In the revised submission, we have done our utmost to address each point. We have focused our efforts on two key major issues, namely the conclusions being drawn from how the HCA model was developed and the experimental validation of some of the key model outputs. As a result, we believe the revised submission is much clearer and stronger. We address each of the reviewer's comments point by point in blue text below.

Reviewer #1 (Remarks to the Author): Expert in mathematical modeling and blood cancers

Multiple myeloma is an osteolytic plasma cell malignancy. The authors propose a hybrid cellular automaton model (agent based model coupled with reaction diffusion equations) describing the interaction of multiple myeloma cells with cells responsible for bone formation and degradation (active & precursor osteoblasts, active & precursor osteoclasts, mesenchymal stem cells) and prototypical signaling/growth factors (RANKL, TGF-beta). The authors show simulation results supporting that the parameterized model is able to recapitulate bone homeostasis. Based on the proposed 2d spatio-temporal model the authors simulate the impact of environment mediated drug resistance (EMDR) and mutation-induced drug resistance on minimal residual disease, treatment failure and myeloma cell heterogeneity.

The proposed manuscript contributes to hypothesis generation, however, does not provide definitive insights into the mechanisms of resistance and relapse. I acknowledge that the literature research, the implementation of the model and its parametrization is a merit in its own right. Nevertheless, key findings presented in this manuscript are straightforward consequences of the model design and the model assumptions. As such the new biological insights are limited. A careful experimental or clinical validation of model predictions is not undertaken. Key results are qualitative without direct implications on the development of novel therapeutic concepts. Taking into account the high complexity of the model, the qualitative nature and the simplicity of the obtained results limit my enthusiasm for the proposed work.

I suggest to further elaborate on the following aspects:

(A) Important model assumptions are not well motivated, some model assumptions seem artificial and do, in my opinion, not reflect plausible biological processes.

- "We assume that each aOC resorbs bone in the direction with the maximum amount of bone." How can cells sense the direction with the maximum amount of bone? What is the biological mechanism or experimental observation underlying this assumption?

This is an important critique and in our original submission we should have done a better job of clearly stating our assumptions and providing the experimental data/published literature that gave us the rationale to make those assumptions. For this particular point, we propose, based on the literature, that in general, osteoclasts resorb bone tissue in areas where there is a high degree of mechanical stress or strain^{1,2}. This is because bone tissue is constantly being remodeled and adapted to changes in mechanical loading, and areas of high stress require more frequent remodeling to maintain their structural integrity². For example, research has suggested that integrins may play a role in guiding the direction of bone resorption by regulating the formation and orientation of the osteoclast sealing zone. Specifically, integrins on the leading edge of the osteoclast are thought to sense the mechanical properties of the bone tissue and orient the osteoclast towards areas of higher mechanical stress or strain^{1,2}. This results in the formation of a sealing zone that is oriented in the direction of the mechanical stress or strain, allowing the osteoclast to efficiently remove bone tissue from areas that are under the greatest mechanical load.

Another hypothesis is that osteoclasts may be guided by the orientation of collagen fibers in bone. Collagen is the main structural protein in bone, and it forms a network of fibers that give bone its strength and resilience. Studies indicate that osteoclasts sense the orientation of collagen fibers in bone through a range of receptors and signaling pathways, and that this information may guide their resorption activity³.

In summary, osteoclast motility and directionality are not random processes but are guided by a multitude of factors and particularly by mechanical stress and strain as evidenced in the literature. We used this information to have osteoclasts resorb in the direction of the highest density of bone since (i) we are unable to model mechanical loading in a static model of bone, (ii) in order to ensure the maximum amount of bone is resorbed in a single osteoclast's lifespan and (iii) introduction of more mechanisms such as collagen directionality, presence of integrins and chemical gradients makes the model too computationally expensive for the purpose of our studies.

Importantly, we have included a clearer rationale in the revised submission, specifically in the supplementary methods, **page 5**, to support our assumptions.

- "In the model, we assume that aOB death is proportional to the amount of bone that was resorbed by the aOC subunit that had occupied the space prior to the aOB (aOC_depth). Therefore, aOB death is also proportional to the average time it takes for an aOB to form one unit of bone (basal_time). This assumption permits the coupling of bone resorption with bone formation, since the lifespan of an aOB controls how much bone the aOB forms. Thus, bone homeostasis is maintained even when the amount of bone resorbed by an aOC varies."

This assumption seems rather artificial to me. How can aOBs sense how much bone aOCs have resorbed? I have the feeling that this ad hoc assumption is put forward to achieve realistic model behavior without having to care about more complicated regulatory mechanisms. However, this makes it very difficult to appreciate and interpret

the properties of the proposed model. This assumption should be discussed in more detail and should be mentioned in the main text.

Bone resorption and bone formation are tightly coupled processes⁴. The definitive mechanisms through which osteoclasts and osteoblasts sense each other and their catabolic/anabolic processes have still not been fully elucidated and can be context dependent. However, several mechanisms are known to play a role including the release of bone derived factors (BDFs; namely TGF-beta and IGF-1), osteoclast derived factors such as sphingosine 1-phosphate, extracellular vesicles, apoptotic bodies containing RANK, and demineralized collagen remnants in the resorption lacunae. To model each of these mechanisms independently would make our model very complex as well as computationally expensive. We therefore integrated a straightforward coupling of OBLs and OCs in the model. Importantly, per the reviewer's comment, we have added a section in the supplementary methods, **page 7**, to make the rationale supporting this assumption much clearer for potential readers and also clarified the associated main text (**page 24-25**)

-"When an aOB is buried, this has possible consequences for bone homeostasis since the aOB did not form as much bone as it would have had it completed its full lifespan. To maintain a steady state of bone, the lifespan of the nearest aOB is increased to allow it to build more bone to compensate for the amount lost due to the buried aOB. This is a simplifying assumption of the model since biologically there are likely to be many factors involved to regulate the number and lifespan of osteoblasts."

This assumption seems rather artificial to me. How can aOBs know that an adjacent OB was buried? I am unaware of biological mechanisms that could explain such dynamics. Even if this assumption is required to obtain reasonable model dynamics, it weakens the mechanistic insights the model can provide. The necessity of such ad hoc assumptions might suggest that important biological mechanisms have not been sufficiently resolved.

There are many unknowns regarding the differentiation process of osteoblasts to osteocytes. For example, it remains unclear whether the decision for an osteoblast to become an osteocyte is dictated by a specific pattern of gene expression in a subset of osteoblasts, a cell autonomous response or, quorum sensing. However, we do have a clearer understanding as to how the communication between osteoblasts and osteocytes occurs and that largely focuses on signaling through gap junctions, formed in large part by connexins. Studies have shown that osteoblasts express multiple connexin isoforms, including connexin 43 (Cx43), connexin 45 (Cx45), and connexin 46 (Cx46). These connexins form gap junctions between adjacent osteoblasts, allowing for the exchange of small molecules and signaling molecules such as cyclic AMP (cAMP), inositol triphosphate (IP3), and calcium ions⁵. Through these gap junctions, osteoblasts can communicate with each other and coordinate their activities to ensure that new bone tissue is deposited in the correct location and in the appropriate amount. This can help to maintain the structural integrity and function of the bone tissue despite the loss of an adjacent osteoblast.

Several studies have investigated the role of gap junctions in osteoblast function and bone formation. For example, studies using mouse models with targeted deletion of connexin 43 have shown that loss of Cx43 in osteoblasts leads to reduced bone mass and altered bone architecture, suggesting that gap junction-mediated communication between osteoblasts is critical for normal bone development and maintenance⁶⁻⁹

Overall, the role of cell-cell communication in osteoblast function and bone formation is complex, however, we do know that bone formation will not occur until a critical number of osteoblasts have formed in the resorption pit and this is sufficient (and necessary) to maintain normal bone homeostasis. Thus, to ensure simplicity, we chose to forgo the inclusion of signaling molecules between osteoblasts and only include the assumption that osteoblasts can communicate with adjacent cells, as evidenced by the literature⁵⁻⁹. To clarify this for potential readers, we have added additional points and references to main (**page 25**) and supplemental text (**page 7**).

- "After an aOC fuses, an MSC is placed on the grid within a certain radius of the aOC (MSC_radius), provided there is not already an MSC within the neighborhood. This requirement ensures that MSCs are located adjacent to sites of bone remodeling to couple bone resorption with bone formation¹⁵. However, to preserve that approximately 0.01% of the bone marrow consists of MSCs², MSCs that are not within a certain radius of an aOC are removed from the grid. Therefore, the number of MSCs fluctuates over time but is capable of returning to the initial condition."

This assumption seems rather artificial to me. Why is the spatial distribution of MSC and their co-localization with aOC not modeled based on first principles?

MSCs have been identified in various niches within the bone marrow. Some of the most common locations where MSCs are found in the bone marrow are the endosteal region, the perivascular region, and the central marrow region^{10,11}. MSCs are located in close proximity to the endosteal surface, where bone remodeling takes place, and the majority of MSCs are in contact with blood vessels or surrounded by pericytes and other undefined cells, which suggests that these cells are located in the perivascular niche^{10,12}. A more recent study used single-cell RNA sequencing to analyze the heterogeneity of MSCs in the bone marrow and identified four distinct populations of MSCs¹³. Regardless of the sub-population of MSC or the niche that it arose from, we know that MSCs are ultimately found near the site of bone remodeling and migrate to areas of high concentration bone derived factors such as upon detection of bone derived factors such as TGF-beta and IGF-1. Thus, we chose to model the aspects of MSCs that are known, rather than adding in complex and computationally expensive additional mechanisms to the math model. We revised the manuscript accordingly, in the supplementary methods, **page 5**, to make this rationale clearer for readers.

- The used mutation rates have to be motivated in more detail. How do the authors know at which rate mutations conferring BTZ resistance occur? How do they know that the considered orders of magnitude are biologically feasible?

This is a valid criticism since our results highlight the importance of the mutation rate in increasing intra tumor heterogeneity. We believe that one of the main advantages of our integrated approach lies in the power of the *in silico* model to explore ranges of parameters that are extremely varied in patients or unknown altogether. That being said, we used mutation rates, now referred to as resistance rates, that are based on the current literature, in which mammalian cancer cells have a probability of developing resistance to numerous treatments of between 10^{-3} to 10^{-6} per cell division^{14,15} regardless of the mechanism of resistance. We have updated the main text of the manuscript, **page 13**, to better reflect this and thank the reviewer for highlighting it.

(B) In my opinion, many model features and key results are straightforward conclusions of model assumptions (the authors get what they put in). Key dynamic features are not an emerging property which is based on more complex regulatory mechanisms but they are more or less directly prescribed. In this sense the first three sections of the results part provide only very limited biological insights. Their main merit (and from the modeling perspective this is a merit!) is to show that the given sets of assumptions and model parameters lead to a plausible model behavior. Also the observations described in the other sections of the results part follow more or less directly from prescribed model features.

We thank the reviewer for this comment. The main goal of our current manuscript is to build a complex multiscale agent-based model of normal bone homeostasis, myeloma growth and its response to treatment. Our model does not try to capture every relevant scale in MM and thus it abstracts the phenotypic impact of those key regulatory mechanisms that the reviewer rightfully mentions. While the link between the genetic and regulatory scales is hardcoded into the model, the result at the tumor and tissue scale that emerge is the result of the complex interplay between the several cellular populations and the microenvironment in ways that are not easy to intuit. For example, perturbing TGF-beta levels in normal bone homeostasis impacts each cell population within the *in silico* model both directly and indirectly. We also would like to state that it took several iterations to get the HCA model to where it was performing as expected but for clarity and space did not include these data. Another major advantage of building this model is that it allows a high degree of temporal resolution, i.e., we can look at the behavior of each population, and their spatial localization, at any time point during normal bone remodeling or subsequent to the incorporation of myeloma. These fluctuations can be difficult to capture biologically and thus the model provides further insights as to cellular dynamics over time that in turn could drive novel hypotheses.

- "Our results show that without EMDR, BTZ completely eradicated the disease at high doses (Fig. 6D, Supplementary Fig. 3j), consistent with our *in vitro* findings. However, when EMDR effects are included in the model, BTZ reduced MM burden significantly, but the disease persisted at a stable, albeit lower, level and did not grow over time compared to existing *in vivo* data (Fig. 6C-d)."

In my opinion, this finding is a direct consequence of the assumptions and the way the authors model EMDR (Supplement: "EMDR is when myeloma cells are transiently protected from Bortezomib due to factors or interactions with other cells in the bone

microenvironment, which we define in our model to be when bone derived factors are above a certain threshold or MSCs or pOBs are within a certain radius of a myeloma cell 21").

In short, the authors assume that a certain percentage of MM cells does not respond to treatment (EMDR). Then they present simulations showing that under this assumption a certain fraction of the cells persists during treatment. Therefore, this observation does not yield new mechanistic insights, it just demonstrates that in a model where it is assumed that not all cells respond to BTZ, simulations show that not all cells respond to BTZ.

- "In the model, the recruitment of MSCs in silico increases over time (Fig 5a) as we and others have previously reported."

This is, I think, the direct consequence of the assumption that MSC increase proportionally to MM cells (Supplement, p. 9 *"We assume that one MSC is recruited per 50 myeloma cells, and that once it is recruited, an MSC remains in the bone marrow"*). As such these results is a double check of model implementation but not more. It basically shows that the simulation evolves according to the underlying assumptions. A similar reasoning may apply to findings 2 and 3 on p. 8 of the main text.

We agree with the reviewer that these results are a double check of model implementation and that simulations show dynamics that emerge from our implementation of the assumptions, particularly that MSCs and OCs increase and OB decrease. However, it is important to note that these complex regulatory networks are working collectively in complex non-linear ways and the results are consistent with our experimental in vivo data and previously published literature, i.e. that the key result is that the model can integrate all of the assumptions made from the literature and our own *in vivo* data and recapitulate how MM progresses in bone. Moreover, we did not assume in the model that preosteoblasts (pOBs) would accumulate at early time points, however, the model faithfully recapitulates the increase and then subsequent loss of pOBs as seen experimentally (**Fig. 5c and 5g**).

- "Similarly, lowering TGF- β levels in silico led to a dramatic increase in bone volume (Supplementary Fig. 2b-e). Interestingly, the HCA results showed that when TGF- β levels dropped below 90%, there was significant bone loss (Supplementary Fig. 2b-e) due to abrogation of MSC proliferation leading to reduced numbers of preosteoblasts that could contribute to bone formation (Supplementary Fig. 2e). These data are reflective of the known biphasic roles of TGF- β on cell behaviors and processes at high and low concentrations", "As expected, BDF has a biphasic effect on bone in which low BDF results in bone loss due to lack of MSC/pOB proliferation and medium BDF results in bone growth due to increased mineralization"

These findings are, in my opinion, rather direct consequences of model assumptions and not properties emerging from first principles or from complex regulatory loops. The assumptions are:

++ Supplement p. 5 *"We assume that high levels of BDF promote proliferation of pOBs whereas low levels promote differentiation 16"*

++ Supplement p. 5 *"We assume that MSCs divide only when BDF is above a certain threshold"*

++ Supplement pp. 6-7 *"This data shows how the mineralization rate compares to the control, specifically that it increases as TGF- β decreases (Supplementary Fig.2a)."*

In short, the authors ASSUME that decreasing TGF-beta levels increase bone formation and that very low levels deplete the cell types responsible for bone formation. Then the authors OBSERVE in their simulations low TGF-beta levels lead to increased bone formation and that very low TGF-beta values lead to practically no bone formation.

"showing that ZOL targets actively resorbing osteoclasts as demonstrated by a reduction in bone resorption per OC (Supplementary Fig. 4h) while BTZ limits the fusion of OC precursors as shown by a reduction in cumulative osteoclast numbers with treatment (Supplementary Fig. 3g)." Also this is, in my opinion, a direct consequence of the assumptions.

We address these comments together since they focus on similar concerns. We totally understand the reviewer's comments that some of the outputs are based on assumptions. We aimed to keep the model simple and demonstrate that normal bone homeostasis, when perturbed, would behave as expected based on experimental and well-established findings in the literature. However, we submit that the key result here is demonstrating that a complex mathematical model can integrate the behaviors and responses of multiple cell types and stimuli and importantly return to homeostasis. For example, we have carefully integrated the biphasic roles of TGF-beta into our model and agree that we observe what we expect based on concentrations, but it is important to note that the mathematical model is using the thresholds of TGF-beta to predict the behavior of multiple cell populations at the same time and that those cell populations are working together to achieve the "observed" results. As we mentioned, our *in silico* model is multiscale with highly coupled cellular populations impacting each other and it took several iterations of the model to achieve one that reflects the known biology. This then gave us confidence in the robustness of the obtained results, with respect to the role of EMDR in driving myeloma heterogeneity which we have now validated biologically in this revised submission (**Fig. 8a-c**). We have also updated the relevant results section, **page 9**, as well to provide more clarity to potential readers.

"We then repeated our in-silico experiments and observed that with the lower mutation probability ($p_{mut} = 10^{-4}$), EMDR increased the likelihood of tumor progression with more simulations likely to relapse (54.4%, $n = 125$) compared to tumors without EMDR (6.4%, $n = 125$; Fig. 6e and g; Supplementary Video 3). Similarly, with a higher mutation probability ($p_{mut} = 10^{-3}$) tumors were more likely to relapse when EMDR is present (100%, $n = 25$) compared to tumors without EMDR (56%, $n = 25$, Fig. 6f and h; Supplementary Video 4)."

In my opinion, it is not surprising that an increase of mutations conferring drug resistance increases the risk of relapsing disease. It is furthermore, I think, not surprising that relapse becomes more frequent if the number of cells protected from treatment (EMDR) and at risk of acquiring resistance mutations increases.

We agree it is not surprising that EMDR protection can lead to frequent relapse but we do think a key finding is that EMDR contributes greatly to highly heterogeneous resistant disease and that blocking the EMDR effect reduces the degree of heterogeneity. This we believe is something that is not well established in the literature and represents a key finding of the HCA model. Importantly, in the revised submission, we have biologically validated this observation *in vitro* and *in vivo* and show that early intervention with zoledronate can block the BDFs being generated to deepen the response to proteasome inhibition with bortezomib and eradicate more treatment sensitive clones that have the potential to mutate and become proteasome inhibitor resistant (See **Figs. 7 and 8**). We believe moving forward that the model we have generated herein will be highly valuable in designing adaptive therapy trials and/or promoting the use of EMDR targeting therapies at earlier time points to deepen MM patient's response to standard of care therapy.

- *"We observed that at a high mutation probability ($p_{mut} = 10^{-3}$) EMDR drove the evolution of significantly more resistant subclones (mean = 5.2 subclones with >10 cells, $n = 24$ simulations) compared to non-EMDR conditions where the arising population was composed largely of a single resistant clone (mean = 1.2 subclones with >10 cells, $n = 10$ simulations; Fig. 7c-d), with similar results at lower mutation probabilities and pulsed BTZ treatment (Supplementary Fig. 9b-c). Moreover, the HCA revealed a significant proportion of resistant subclones originated in close proximity to MSCs/pOBs (Fig. 7c) or following release of BDF from osteoclastic bone resorption (Supplementary Fig. 9c). Taken together, these data would indicate that EMDR not only leads to enhanced relapse, but increased MM heterogeneity due to the emergence of independent resistant MM clones." As described above this is, in my opinion, a direct consequence of the model assumptions.*

Our updated manuscript now includes *in vivo* evidence showing that targeting osteoclasts with zoledronate to minimize BDF availability improves the effectiveness of BTZ in eliminating drug sensitive cells (**Fig. 6-8**). These data support the predictions made by the model and provide a clearer picture to the benefits of applying bone/EMDR targeting agents for the treatment of MM to deepen therapeutic responses. This approach did not eliminate the protective effects offered by MSCs and we are currently examining the distance between surviving drug sensitive MM cells and MSCs in the BTZ plus ZOL treated cohort. These data will be integrated into the model in ongoing studies and our future work will examine if blocking EMDR provided by MSCs (IL-6 neutralizing antibodies for example) can again enhance the efficacy of BTZ.

- *"Our data demonstrates that resistant disease cannot develop without MM intrinsic mechanisms."*

I think this conclusion is straightforward: If treatment resistance is restricted to limited niches such as in the concept of EMDR, there can be no unbounded expansion of cells in presence of treatment unless cells acquire additional (EMDR-independent) resistance mechanisms.

We agree that our conclusion would seem obvious under continuous treatment as sensitive cells are always exposed to drug and can have no unbounded expansion of

cells. However, we also demonstrate this with pulsed treatment (**Supp Fig. 8-9**) and no probability of developing resistance (**Fig. 6d and 6g**). Pulsed treatment more accurately reflects how bortezomib is delivered to patients, whereas continuous treatment better reflects published animal studies¹⁶. The advantage of our mathematical model is that we can apply treatments and their timing in any number of ways. Using this pulsed approach in our model, during times where treatment is off, sensitive cells are still able to grow and proliferate. This growth leads to more protective MSCs/preosteoblasts and more BDFs from increased osteoclastic bone resorption, however, when treatment resumes, MM cell growth is reduced. Over time, pulsed therapy leads to increased MM growth, but it is not sufficient for relapse to occur. This only happens when MM cells gain intrinsic resistance (**Supp fig 8**).

(C) Results have to be formulated in a much more decent and careful way.

"For example, we show that the bone ecosystem contributes to minimal residual disease, and MM clonal heterogeneity" This is in my opinion not fully true. The authors propose a hypothetical mechanism by which the micro-environment protects MM cells. Simulations show that such a mechanism (straightforwardly) leads to more MRD and MM clonal heterogeneity. However, the authors provide no compelling evidence for the existence of such a mechanism in vivo. E.g., they do not show the existence of BTZ sensitive cells in BTZ refractory patients.

The reviewer makes an important point that we have done our utmost to address in this revised submission. Currently, there is no single clinically relevant PI resistance marker available making it difficult to address this critique in human tissue microarrays for example. We therefore used two approaches;

We availed of the Moffitt *ex vivo* mathematical myeloma advisor (EMMA) platform¹⁷. This technology allows for the testing of reagents as single therapies or in combination, on patient derived CD138+ selected MM cells cultured in the presence of patient-derived bone marrow stroma over a 6-day period using live cell imaging. An area under the curve (AUC) up to 96 hours for each dose is calculated and results are displayed as a mean AUC. In CD138 positive samples derived from patients considered to be refractory to PI, our analysis shows that approximately 20% of patients are still capable of responding to BTZ treatment *ex vivo* (**Fig 7g-h**). This suggests that there is a reservoir of PI sensitive cells in PI refractory patients and supports the idea that the bone microenvironment can protect drug sensitive cells from chemo or targeted therapies. *In vitro*, we show that the co-culture of PI sensitive MM cells with mesenchymal stem cells offers significantly more protection from BTZ as predicted by the mathematical model (**Fig 7**).

In vivo, we show that bisphosphonates - which prevent the release of BDFs by killing osteoclasts- reduces the number of safety zones for MM sensitive cells leading to a deeper response to BTZ and a rapid outgrowth of BTZ resistant cells (**Fig. 7-8**).

Taken together, these data validate key model predictions. We do fully appreciate the reviewer's concerns, however, and have made sure that the conclusions drawn are supported as much as possible by experimental data.

- "*The similarities in trends between the in vivo model and the HCA outputs, indicated that the HCA model was accurately recapitulating the MM-bone vicious cycle.*"

The authors come to the conclusion that the HCA model accurately recapitulates the MM-bone vicious cycle. However, there exist substantial differences between the model dynamics and experimental results shown in Fig 4: For most of the time the increase of MM/MA (panel b) shows an approximate linear increase in the simulations, whereas in the experiments (e) the increase looks clearly exponential. The same applies to the decay of BA/TA shown in panels (d) and (e). The dynamics between 0 and 30 days also look quite different in panels (c) and (f). I have the feeling that the authors may miss some mechanisms governing the dynamics of the MM cell expansion.

- Also in Figure 5 (pairs d,f and b,h) the simulations seem to differ qualitatively from the experimental data

It is true that models based on on-lattice CAs such as ours do not capture exponential growth. The technical explanation is that, as opposed to real cells, our cells can only divide if they are immediately adjacent to empty space so while in exponential growth every cell can divide every cycle, in our model it is mainly those on the boundary that do. That is one of the key reasons why, when we compare computational and experimental data, we focus more on the qualitative trends than on quantitative cell number to cell number comparisons. Other agent-based models such as off lattice ones allow for cells to proliferate potentially every time step but they also present problems: 1) they are computationally more demanding which would complicate running the number of simulations we performed for this research and 2) they allow for cells to divide even when surrounded by many cells which we know, biophysically, is not possible either.

Furthermore, biological outputs typically only provide us a snapshot of what is happening at a given time point in a given animal. In contrast, the power of the *in silico* model is that it allows us to visualize and quantify what is happening at any given time in any one simulation. Here, the amount of data points is limited in the biological data (n=5 per time point). However, in the computational data, we are able to quantify outputs at any time step and thus have a much more detailed output. Importantly, we are able to match the following populations:

- MSCs – a dramatic increase in MSC content in MM tumors. (**Fig. 5a and e**)
- pOBs – an early increase in pOBs followed by a complete loss. (**Fig 5c and 5g**)
- OBs – MM dependent loss of OBs (**Fig 5b and 5f**)
- OCs – MM dependent increase in OCs (**Fig 5d-h**)– it was interesting to observe in the *in silico* model that OCs decrease eventually, however, we did not see this in vivo and this may occur at later time points when all bone is lost.

We have ensured any conclusions we made are supported by the presented data.

- “When EMDR is absent, many tumors go extinct and relapse only occurs if intrinsic resistance arises early following treatment, whereas with EMDR, sensitive cells persist and resistance can arise later in the course of the disease. Taken together, these results demonstrate the ability of the HCA to predict the effect of mutation probability and EMDR on tumor relapse and indicate a vital role of EMDR in contributing to minimal residual disease and the development of relapse refractory MM.” This is a hypothetical finding which has not been validated experimentally.

To validate these findings experimentally, we utilized a co-culture consisting of nuclear eYFP labelled U266 (Nuc-eYFP U266) and human MSCs (Fig. 6j-k). Nuc-eYFP U266 were cultured together with MSCs or alone to represent the presence and absence of EMDR, respectively. The U266/MSc and U266 only cultures were then exposed to vehicle or high dose of bortezomib for 72 hours, where the Nuc-eYFP U266 confluency was quantified by live cell imaging using the Incucyte SX5 system. After this period, treatments were removed and replaced with fresh media every 3-4 days for a period of 3 weeks. Nuc-eYFP U266 confluency was quantified every 3-7 days by live cell imaging with the EVOS auto system. Our data demonstrates that without the presence of MSCs, Nuc-eYFP U266 MM cells die. However, addition of MSCs, leads to a proportion of cells remaining, roughly one third of the initial number of cells. These data, included now as Fig. 6, validate our *in silico* findings, that EMDR protects MM cells and leading to minimal residual disease.

Further, whilst we are unable to modify the rate at which MM cells become resistant, we included further co-culture experiments where we included varied numbers of PI-resistant MM cells (PSR-RFP). We observed that following treatment with bortezomib, total MM numbers (U266 + PSR) initially reduced as expected but ultimately the resistant cells take over when MSCs are not present. However, as we hypothesized from our *in silico* predictions, without EMDR/MSCs, these wells of MM cells were entirely PSR-RFP. When MSCs were present, this protected several sensitive Nuc-eYFP U266 thus validating our predictions and supporting the role of adaptive therapy as a potential treatment for MM (Fig 6.). The role of adaptive therapy in MM is the focus of our ongoing studies

(D) Key conclusions are qualitative, some of them not new. The authors do not use the full potential of their model.

- The authors state “A major additional advantage of our model is its spatial nature, allowing us to visualize and measure where the ecosystem, and particularly stromal cells, contribute to the generation of resistant clones.” However, they never compare the 2d simulations to 2d data. Why?

There is no singular definitive marker for PI-resistance, therefore, it is difficult to localize the proximity of newly PI-resistant cells to stroma histologically. However, to address this, we injected mice with GFP+ PI sensitive cells or RFP+ resistant cells allowing us to compare *in vivo* spatial outputs to those generated by the HCA model (Fig. 8d). We have begun quantifying the spatial relationship of surviving GFP+ cells in the BTZ and ZOL+BTZ treatment groups to the bone surface and MSCs. Here we show that in BTZ treated mice (Green histogram), surviving GFP+ cells remain closer to the

bone surface (<300 microns) than compared with ZOL+BTZ treated mice (blue histogram). In addition, we are continuing to quantify same output for MSCs and surviving GFP+ MM cells. However, due to discontinuation of the antibody we originally used, it has been necessary to use alternative antibodies which we feel need to be better optimized so we can be sure our results are rigorous. Thus, we feel this data is too preliminary to add to our current manuscript and will be used in future iterations of the model.

- the authors use population-based quantities to compare model simulation to experiments (Fig 4 & 5), although the model and the histology are 2d. To infer population based quantities the spatial model may be over-complicated and bearing the risk of over-fitting.

We believe this approach is justified, i.e., where we have matched model outputs and experimental data qualitatively rather than quantitatively and are looking at the model replicating trends.

"Together, these data support that early targeting of bone marrow microenvironment, and thus EMDR, would lead to reduced MM heterogeneity and potentially deeper responses with subsequent lines of treatments." "In conclusion, our data demonstrates a significant role for the bone ecosystem in MM survival and resistance, and suggests that early intervention with bone ecosystem targeting therapies may prevent the emergence of heterogeneous drug resistant MM."

This conclusion is not new. See. e.g., <https://www.nature.com/articles/s41467-020-19932-1>

Kawano Y, Moschetta M, Manier S, Glavey S, Görgün GT, Roccaro AM, Anderson KC, Ghobrial IM. Targeting the bone marrow microenvironment in multiple myeloma. *Immunol Rev.* 2015 Jan;263(1):160-72. doi: 10.1111/imr.12233. PMID: 25510276.

From the side of drug-development the challenge of MM therapy is not a shortage of concepts. The main complication is to identify the proper compounds and a feasible way to deliver them (see. e.g., Federico C, Alhallak K, Sun J, Duncan K, Azab F, Sudlow GP, de la Puente P, Muz B, Kapoor V, Zhang L, Yuan F, Markovic M, Kotsybar J, Wasden K, Guenther N, Gurley S, King J, Kohlen D, Salama NN, Thotala D, Hallahan DE, Vij R, DiPersio JF, Achilefu S, Azab AK. Tumor microenvironment-targeted nanoparticles loaded with bortezomib and ROCK inhibitor improve efficacy in multiple myeloma. *Nat Commun.* 2020 Nov 27;11(1):6037. doi: 10.1038/s41467-020-19932-1. PMID: 33247158; PMCID: PMC7699624.) These issues are, unfortunately, not at all addressed

Whilst we agree that our conclusion of EMDR playing an important role in multiple myeloma is not new we do believe many of the outputs are novel, in particular in regard to the role EMDR plays in driving heterogeneous resistant disease. Further the primary focus of this study was to develop an integrated in-silico agent based model of myeloma that recapitulates how multiple populations interact in concert with each other to promote disease progression. Importantly, the model does this in a spatial manner.

We believe our results pertaining to EMDR are particularly interesting. Resistant MM cells are often cross-resistant to other drugs, for example the PSR cell line is also resistant to carfilzomib, ixazomib, doxorubicin and melphalan (*unpublished observation*). Many of these EMDR-targeting trials are performed in RRMM patients. Our data demonstrates that when resistance has already developed, EMDR inhibition is no longer beneficial (**Supp Fig. 10**). These findings strengthen the need for EMDR targeted agents to be used early in treatment before resistant cells become the dominant MM population which we believe is a novel finding.

Many of the 'EMDR-targeted' agents that have ultimately failed in clinical trials only inhibit one aspect of EMDR, for example IL6 from MSC (Siltuximab), MM-MSK interactions (AMD3100) or bone-derived factors (Zoledronate). Our data supports the fact that multiple niches are responsible for EMDR and that it may be necessary to target more than one of these EMDR mechanisms at the same time to achieve the best outcome. Our data provides rationale for targeting EMDR from multiple angles in order to deliver the maximum effect. That being said, as the reviewer commented, it is still important to elucidate the exact mechanisms which protect MM cells through MM treatment such that novel targeted agents can be developed and we have added to the discussion to clarify this. Identifying which components of BDF and which MSC-derived factors are responsible for EMDR is the topic of an on-going studies.

(E) The sub-model of bone remodeling has to be delineated from previous models by the same authors. What is actually new?

Our new HCA model of the bone cancer microenvironment was developed from scratch but based on our own experience developing agent-based models of the bone that include ecosystem cells such as osteoclasts and osteoblasts. Similarities between our previous model and the current one is our emphasis on studying cancer progression developing from an existing bone homeostasis resulting from the interactions between osteoclasts and osteoblasts and other related cellular species. Differences include (but are not limited to):

- Code written from scratch in both cases and calibrated against different (literature parameters in the first model and our own for the one in this manuscript) experimental data. This makes this model significantly more robust and reliable.
- Cancer cells are parameterized with data from prostate cancer in the old model while using myeloma in the new one to capture key differences between osteolytic and osteoblastic disease that emerge naturally in both models.
- The new model is parameterized using more of our own experimental data instead of being parameterized using mostly published literature with only a few key experimental data as is the case in the old model.
- The new model is more robust and capable of performing bone remodeling simultaneously in several sites. This robustness is also demonstrated by the fact that homeostasis in the bone is preserved even with the use of variable depths of osteoclast-mediated bone resorption which was not the case previously.
- The new model includes more biology that is key in regulating bone remodeling such as the role of osteocytes and bone-lining cells in driving the initiation of bone remodeling.
- New model code was developed using HAL¹⁸ for modularity, extensibility, reusability by other researchers.

In brief, although there are similarities given that both models are ABMs that capture bone homeostasis and homeostasis disruption as a playing ground for cancer, they have been parameterized and calibrated very differently, they have been designed to study diverse types of cancer and cancer treatments and to address different questions about the nature of treatment resistance. We have included these points in the revised submission (**Supplementary, page 2**) and thank the reviewer for this important comment.

(F) I have the feeling that the authors did not invest sufficient efforts to validate the predictions of their model experimentally

- According to the model predictions (e.g., Fig 7b) BTZ sensitive MM cells should be detectable in BTZ refractory individuals. Why didn't the authors check this?

This is a really important point and we have taken the time to show the presence of BTZ sensitive cells in PI-refractory patients. Please see response to point (C) above and **Fig 7g-h**.

- the authors did not undertake gene sequencing (e.g., from patients at different disease stages) to validate their predictions of clonal heterogeneity.

We appreciate the reviewer's comment, however, the idea of clonal heterogeneity and clonal evolution is a well-known concept in multiple myeloma^{19,20}. Our results thus far

point to a significant role of EMDR in the development of clonal evolution and heterogeneity. Currently, there are no available drugs that solely target EMDR approved for the treatment of MM patients. For this reason, performing single cell sequencing to identify clonal heterogeneity on MM patients would only serve to validate an already known concept, in that clonal heterogeneity is present. Moreover, it would not validate our findings that EMDR contributes to heterogeneity as EMDR will not have been inhibited in any patients.

(G) Some of the experimental setups require a more detailed discussion:

- "*Based on our in vitro observations, we also incorporated a cost of resistance in the model such that BTZ-resistant cells proliferate at a slower rate and are out-competed by PI-sensitive cells in the absence of treatment*"

Can the authors be sure that the resistance mutations themselves leads to a competitive disadvantage? In my opinion it cannot be ruled out whether other differences (other mutations, epigenetic differences, metabolic differences) of U266 and PSR cells are responsible for the observed differences in proliferative fitness. Can the in vitro observations be reproduced in competitive transplantation assays? Can the fitness costs be observed also in case of cells harboring only the specific mutations required for resistance (e.g., mutations at drug binding site) but no additional mutations/aberrations? This should be tested e.g., in a CRISPR-based approach.

We agree with the reviewer that we cannot be certain that a specific mutation or alteration is responsible for a competitive disadvantage. Further, it was an oversimplification to state that a mutation is responsible for the observed resistance phenotype when in fact, as reviewers 1 and 3 pointed out, resistance may arise from mutations, epigenetic differences, metabolic differences or alterations in signaling pathways. For this reason, we have changed the term p_{mut} to p_{Ω} (probability of resistance) as the term is really the probability that a cell will develop a resistance phenotype, regardless of the mechanism.

Further, as mentioned previously, MM is a vastly heterogeneous disease with few if any clinically targetable resistance mechanisms identified. As such, many labs around the world, including our own, focus on identifying novel, tractable and targetable resistance mechanisms for the treatment of relapsed-refractory multiple myeloma. To identify and validate a specific and clinically relevant mechanism of resistance for MM PI-resistance is outside the scope of this manuscript. Moreover, the *in silico* model is not fixated on specific resistance mechanisms currently but more on whether a cell is sensitive or not. One such benefit of the *in silico* model is the ability to alter parameters, such as p_{Ω} , response to treatment and doubling time, to mimic the heterogeneity seen in the clinic. In **Supplementary Fig. S10**, we demonstrate this ability and how modulating different parameters alters disease outcomes. We have added more discussion on these points as requested (**Page 22**).

- The authors conduct experiments with TGF-beta. Can they really assume that the observed changes are representative for all bone derived factors?

TGF-beta is one of the most abundant non-collagenous proteins in bone^{21,22} and serves as a good representative bone derived factors since it plays a role in bone development and homeostasis^{23,24} including 1) recruitment and 2) proliferation of MSCs²¹, 3) differentiation to preosteoblasts/mature osteoblasts and 4) mineral deposition (our own experimental data; **Fig. 3-4**). Obviously other bone derived factors can play a part including IGF-1, BMPs, activin-A but for simplicity and the fact that TGF-beta plays such a potent role in skeletal health and cancer progression, we focused on this factor to model. We have also added more explanation on this point to the main text (**page 7**) to clarify it for potential readers.

Minor:

Suppl. Tables:

- Please make sure to describe how each parameter indicated as "Estimated" was estimated. I miss (or overlook) e.g., the explanation for **maxRANKL** or **Ts**

We appreciate the reviewers comment and provide the below information for clarity:

- MaxRANKL was determined by running the unnormalized model a number of times and then looking at the output "pOBborn.csv" to evaluate the maximum value of RANKL that occurred when an aOC formed. This value of RANKL was then set as maxRANKL so that most values of RANKL that lead to the formation of an aOC are between 0 and 1 (however, it is possible that some values in future runs of model are not strictly below 1).
- maxTGFB was determined by running the unnormalized model a number of times and then recording the maximum TGFB at a single grid point. This value of BDF was then set as maxTGFB. The maximum TGFB is printed after each model is run (tmax) so that it can be monitored whether the value stays between 0 and 1.
- Ts is the basal BDF and is defined as written in the table, $\alpha_B / (|\delta_T| \cdot BDF_{max})$. This is the amount of BDF that is always present on the grid.

We have added this information to the supplementary methods, **page 15**.

- It would be nice to see how many data points were used for the fits mentioned in the supplement and main text.

The data points used to fit the exponential decay function are shown in **Supplemental Fig. 2b**. This is also further detailed below.

- Please specify the meaning of the black / red dots in Sfig 2 b c g

We have taken the time to clarify the meanings of black and red dots in Supplementary Fig. 2 as follows:

- Supp Fig. 2b - The black dots represent the data from the *in vitro* experiments with the MC3T3-E1 cells that were used to fit the exponential decay function, as described in section 1.1 of supplementary methods.
- Supp Fig. 2c - The red dot represents the lack of MSC/pOB division rate when BDF is at a basal level (dotted red line), and the black dot represents the division rate when BDF is at the threshold level for proliferation (dotted black line). Another black dot highlights the half maximum division rate defined by the Hill function.

- Supp Fig. 2g: The black dot highlights the half maximum fusion probability defined by the Hill function.

These clarifications have been added to the legend of **Supplementary Figure 2**.

Fig 1b:

- please add a reference to each arrow and add in which experimental system the respective results were obtained.

We thank the reviewer for this insightful suggestion. We have updated **Figure 1b**, in line with reviewer 2's comments. In addition, we have included the figure and table below with references and experimental systems in which respective results were obtained. The addition of references and a table, in our opinion, overcrowds the original figure but we thought it would be beneficial to potential readers to add it as **Supplementary Figure 1a**.

Line number	Experimental system and reference
1	Rat in vivo (PMID: 15704000), Mice in vivo (PMID: 21909105)
2	Human in vitro (PMID: 12958198); (PMID: 15248232)
3	Mouse in vitro (Supplementary Fig. 2f)
4	Mouse in vivo , human in vitro (PMID: 19584867)
5	Mouse in vivo , human in vitro (PMID: 19584867)
6	Mouse in vivo , human in vitro (PMID: 19584867)
7	Mouse in vivo , human in vitro (PMID: 19584867)
8	Mouse in vitro (Supplementary Fig. 2a)
9	Mouse in vivo (PMID: 22729283)
10	Mouse in vivo (PMID: 25079226)
11	Mouse and Human in vitro (Fig. 3f, Fig. 6j-m)
12	Mouse in vivo (Fig 5e)
13	Mouse in vitro and in vivo (PMID: 22102554)
14	Human in vitro (PMID: 15933061)
15	Human/mouse in vitro (Fig. 3f)
16	Human in vitro and in vivo (PMID: 11739153), Mouse in vitro and in vivo (PMID: 11562486)
17	Human in vitro (Fig. 3e) Mouse in vitro (Supplementary Fig. 3b and 4b)

What are the units in equations 6-8?

We thank the reviewer for the insightful question and the opportunity to clarify. Below are the units for the equations 6-8.

- Equation 6:
 - aOB_Death: days
 - basal_time: days per unit of bone
 - aOC_depth: units of bone
- Equation 7:
 - This is unitless since it represents fold change to bone mineralization time
- Equation 8:
 - Mineralization_Time: days per unit of bone
 - basal_time: days per unit of bone

We have updated the supplementary methods, **pages 7-8**, to reflect these changes.

- Equation 6 requires a more concise explanation

This equation describes how to define the lifespan of an aOB so that it lives long enough to replace the bone resorbed by an aOC (during homeostasis). We have updated the supplementary methods, page 7, to reflect these changes.

Reviewer #2 (Remarks to the Author): Expert in mathematical modeling and agent-based modeling

Key results:

Multiple myeloma (MM) is a treatable plasma cell cancer. However, people who suffer from this disease will eventually develop resistance to the various lines of therapy. In this article, the authors investigate the impact of the bone marrow microenvironment on treatment resistance of MM by constructing a mathematical model, a hybrid cellular automaton (HCA), parametrized to in vitro, in vivo, and human data including their own MM mouse model and cell cultures. Interestingly, their model can capture the positive feedback on MM growth caused by the bone marrow environment. Their results suggest that environment mediated drug resistance (EMDR) protects a small number of MM cells from treatment, contributing to residual disease. Furthermore, they found that EMDR promotes heterogeneity in treatment-resistant MM cells, increasing the challenge of implementing efficient relapse therapies.

Validity:

For the most part, the manuscript is quite robust. I understand that the objective of the mathematical model presented is to capture key outputs and not to precisely fit the experimental results. However, I enquire about the criteria to accept the model's results as a valid representation of the phenomenon as some of the simulations are not measured in the same units or span the same time scale. Furthermore, from Fig 3, I wonder if it was tested that the protective effect is entirely due to bone derived factors (BDFs). What happens when those BDFs are inhibited? Will the spatial distribution of the apoptotic and dividing cells change?

Here we provide evidence that by treating myeloma bearing mice with Zol (pink bars), an osteoclast inhibitor and thus an inhibitor of BDF release²¹, we are able to change the spatial distribution of GFP+ MM cells such that MM cells are further away from the bone surface in the Zol treated group compared to vehicle treated mice (gray).

In regards to the fact that the immune component is omitted in the study, one of the representative BDFs is TGF- β , a cytokine with many roles in immune regulation. What can be said about the effects of TGF- β on the immunodeficient mice model used?

This is an important comment given the role of the immune microenvironment in cancer progression, particularly in multiple myeloma which has been responding well to BCMA targeted CAR-T. Obviously TGF beta levels would have a profound effect on immune infiltrates and this is something we plan on integrating into the model as we validate our initial findings *in vivo*. We often use the immunocompetent 5TGM1 model and so are in a position to add key immune players in future versions of the mathematical model. We have added a passage to the discussion to address this point (**Page 21**).

Significance:

The results show that even with the immune system excluded from the study, the bone marrow environment is a non-negligible aspect of treatment resistance in multiple myeloma. Therefore, it further supports the idea of targeting the tumour microenvironment in second-line therapies. They also give us insights on the mechanisms leading to the development of heterogeneity and resistance in multiple myeloma.

Data and methodology:

In the methods, what is the reason for choosing the two types of treatment administration used for the HCA model?

We apologize for not stating this clearly in the initial submission. We chose to model two types of treatment, as frequently in mouse models, treatment is delivered continuously until the end point. However, in patients, treatment is delivered in cycles with periods where treatment is off, which we termed pulsed therapy. We have updated the main text, **page 27**, for clarity.

For the results, in Fig. 7a, I am confused as to why the simulation shows that there is resistant MM with EMDR (grey, resistant MM + EMDR) in the case where there is an absence of EMDR (no EMDR). There are also small details I noticed about the presentation of the figures. In Fig. 1b, the black arrows are not defined. Fig. 4 b-g, Fig. 5 a-d, and Fig. 7b and 7d have missing or incomplete legends. I was not provided the coding portion of the computational algorithm, so I cannot comment on it.

We thank the reviewer for these helpful comments and apologize for the oversight. In the updated manuscript, we have included:

- We apologize for the oversight regarding **Figure 7a**. Resistant MM, in the in-silico model, do not receive EMDR since they are intrinsically resistant to PI treatment. As such, the gray color denotes resistant MM cells that are in close proximity to MSCs and/or BDF. This has been updated in **Figure 7a**.
- **Figure 1b** has been updated with new colors added to arrows and definitions to all. In addition, in line with Reviewer 1's suggestion we have added a more detailed figure to the supplementary which contains references and experimental systems in which effects were observed.
- **Figures 4, 5 and 7** all have updated legends.

- The coding was provided as part of the initial submission as an additional file. We have included the code and the location for download in the code availability statement. It is also available in this repository: https://github.com/dbasanta/MM_ABM

Analytical approach:

Like mentioned above, there is no mention of the method used to compare the results of the HCA model to the experimental data. It appears to be a fully qualitative comparison, and maybe a quantitative test could improve the strength of the analysis.

Thank you for bringing up this important point. We decided to match model outputs and experimental data qualitatively rather than quantitatively for reasons including (1) on-lattice CAs do not capture exponential growth, and (2) there are likely to be some temporal differences between the 2D model and the population-level data such as myeloma growth in the marrow.

Suggested improvements:

I have no suggestion for additional experiments or simulations.

Clarity and context:

The overall clarity and accessibility of the text are quite good, but there are some things to note.

1. A few acronyms should be defined before their use (BCMA, RANKL, CAR, TGF- β) and some can be dropped altogether, such as OC for osteoclasts, to help with the flow in the reading.

We appreciate the reviewer's suggestion and have added additional definitions of certain acronyms (BCMA, RANKL, CAR, TGF- β) and removed some for clarity as suggested.

2. In the research questions found in the introduction, it asks how EMDR and intrinsic drug resistance both contribute to the heterogeneity of the disease in control and treatment conditions. Explicitly, what is the control condition in this situation? Furthermore, it says that the three research questions are difficult to address using current in vivo and in vitro techniques. It would feel more complete to explain and show cited examples as to why.

We feel that in our initial manuscript the control conditions were included. Those being, no EMDR and no ability for MM cells to develop resistance. These control conditions are not possible in biology as A) we cannot prevent cancer cells from undergoing changes that lead to resistance and B) we cannot remove the microenvironment from an animal/patients etc.

3. As the journal is not specifically targeted to people familiar with mathematical models, I think a separate sentence for the description of hybrid cellular automata models, instead of using parentheses, would give more context in the introduction.

We have added an additional sentence to describe the HCA model (page 6).

4. In the HCA model, there is significant bone marrow loss when TGF- β levels are dropped by 90%. This is in comparison to normal physiological values?

Yes. We initially demonstrate that when TGF- β levels fall to 90% of physiological levels then we observe bone loss. This is in comparison to published studies where TGF- β , TGF- β receptors or other downstream signaling pathways have been systemically knocked out or specifically in osteoblasts and their precursors, demonstrate dramatic skeletal defects (summarized in ²⁵ and ²⁴). Similarly, in studies where TGF- β is only partially inhibited, such as with inhibitors ²⁶⁻²⁹ bone levels are increased, as reflected in by our data (60% inhibition; **Supplementary Fig 2d-e**).

5. In the discussion, there is a description of different mathematical models that explore the interaction between MM and its microenvironment for which the objective is not clear. It could be rewritten to explain, for example, if it is to demonstrate the novelty or the importance of the HCA model.

We thank the reviewer for their important point. The purpose of this section was to highlight that other models of the MM and bone marrow microenvironment have been made. However, our model is the only model that has been integrated with biological data and tuned over several iterations. Additionally, many of these models do not include a spatial aspect whereas the HCA model does, thus highlighting another novel aspect of our model choice. We have adjusted this paragraph to clarify our objective (**main text, page 18**).

References:

The manuscript referred to the literature appropriately.

Your expertise:

My expertise is in computational medicine and mathematical modelling; therefore the in vitro and in vivo methods are outside the scope of it.

Reviewer #3 (Remarks to the Author): Expert in multiple myeloma genomics, therapy, and pathogenesis

This is an elegant model of the interaction between myeloma tumor cells and the bone microenvironment, incorporating in vitro and in vivo observations into in silico modeling. It is clinically relevant, particularly given that the model addresses potential mechanisms supporting minimal residual disease persistence.

My comments are minor:

1. I think it is an overstatement to say that MM resistance can't develop without cell intrinsic mechanisms. Your model includes mutations as only mechanism of resistance available- it is not including the role of the immune microenvironment, nor any non-

mutational mechanisms described to be associated with PI- resistance ie changes in signaling pathways / gene expression / metabolism. I would change this phrasing in the abstract.

We agree that the development of resistance is complex and that it is an overstatement to say that mutations are responsible for intrinsic resistance. We have changed the phrasing to discuss all intrinsic mechanisms and not solely mutations throughout the manuscript.

2. For all in vivo studies please include the number of mice in the figure legends.

We have included all N numbers in the figure legends and thank the reviewer for catching this.

3. In Figure 4 it appears that the bone destruction occurs ahead of the tumor expansion. How do you explain this? And how can you say it's myeloma-induced bone loss if the loss is mainly prior to 10% GFP+ cells?

Upon seeding the bone microenvironment, the MM cells frequently home to the trabecular bone and endosteum, thus the MM cells are concentrated in specific regions. However, the flow cytometry data measures all cells in the entirety of the bone. It is plausible that these MM cells, concentrated in specific locations, can produce factors to initiate the vicious cycle before colonization of the entire bone.

4. In Figure 6, you model the effect of bortezomib starting at 10% PC only- how does the model behave if you have a higher % of MM prior to treatment?

This is a great question posed by the reviewer. We spent a long time initially deciding on when we should initiate our treatments. As MM is defined as at least 10% PC in the bone marrow, we settled on this starting point. Further, as the simulation progresses and more and more cells are added to the simulation, each simulation becomes more computationally expensive. Whilst we have not performed experiments at higher than 20% PC content, owing to the computational power required, we can offer our predictions based on what we have observed. For example, we predict that a higher % of MM prior to treatment would lead to a higher probability of relapse and shorter relapse times compared to a lower % of MM cells.

5. Bortezomib-resistant cells do not always proliferate more slowly- what happens in the model if you remove the cost of resistance?

We have observed in matched pairs of MM that resistant cells frequently proliferate more slowly, but as stated, not always. In **supp figure 10**, we have demonstrated how manipulation of this parameter affects the outcomes.

6. You state that early intervention with bone ecosystem targeting therapies may prevent the emergence of heterogenous drug resistant MM- can this be tested in silico in your model?

Yes, in future iterations we aim to apply multiple known "EMDR" targeting agents such as AMD300, anti-IL6 antibodies and ZOL to name a few, alone or in combination with BTZ, in standard treatment schedules or in adaptive therapy manner. We hope to find the best combination and treatment scheduling to delay the onset of heterogeneous resistant disease.

7. Are you able to show data with dexamethasone? Clinically relevant given that it's in most MM regimens, and biologically relevant given that it affects both MM and bone architecture.

Here we primarily use BTZ and ZOL for proof of principle. The beauty of the mathematical model is that it can be easily adapted to integrate the effects of other standard of care therapies as single agents or in combination such as DEX. This is an important point and we have expanded upon it in the revised discussion (**Page 21**).

8. The discussion has a good description of the model limitations; I think it's well written

We thank the reviewer for their kind comments.

References:

- 1 Wang, L., You, X., Zhang, L., Zhang, C. & Zou, W. Mechanical regulation of bone remodeling. *Bone Res* **10**, 16 (2022). <https://doi.org/10.1038/s41413-022-00190-4>
- 2 Novack, D. V. & Faccio, R. Osteoclast motility: putting the brakes on bone resorption. *Ageing Res Rev* **10**, 54-61 (2011). <https://doi.org/10.1016/j.arr.2009.09.005>
- 3 van den Dries, K., Linder, S., Maridonneau-Parini, I. & Poincloux, R. Probing the mechanical landscape - new insights into podosome architecture and mechanics. *J Cell Sci* **132** (2019). <https://doi.org/10.1242/jcs.236828>
- 4 Durdan, M. M., Azaria, R. D. & Weivoda, M. M. Novel insights into the coupling of osteoclasts and resorption to bone formation. *Semin Cell Dev Biol* **123**, 4-13 (2022). <https://doi.org/10.1016/j.semcdb.2021.10.008>
- 5 Plotkin, L. I. & Bellido, T. Beyond gap junctions: Connexin43 and bone cell signaling. *Bone* **52**, 157-166 (2013). <https://doi.org/10.1016/j.bone.2012.09.030>
- 6 Stains, J. P. & Civitelli, R. Cell-cell interactions in regulating osteogenesis and osteoblast function. *Birth Defects Res C Embryo Today* **75**, 72-80 (2005). <https://doi.org/10.1002/bdrc.20034>
- 7 Stains, J. P., Watkins, M. P., Grimston, S. K., Hebert, C. & Civitelli, R. Molecular mechanisms of osteoblast/osteocyte regulation by connexin43. *Calcif Tissue Int* **94**, 55-67 (2014). <https://doi.org/10.1007/s00223-013-9742-6>
- 8 Civitelli, R. Cell-cell communication in the osteoblast/osteocyte lineage. *Arch Biochem Biophys* **473**, 188-192 (2008). <https://doi.org/10.1016/j.abb.2008.04.005>
- 9 Watkins, M. *et al.* Osteoblast connexin43 modulates skeletal architecture by regulating both arms of bone remodeling. *Mol Biol Cell* **22**, 1240-1251 (2011). <https://doi.org/10.1091/mbc.E10-07-0571>
- 10 Bianco, P. *et al.* The meaning, the sense and the significance: translating the science of mesenchymal stem cells into medicine. *Nat Med* **19**, 35-42 (2013). <https://doi.org/10.1038/nm.3028>
- 11 Greenbaum, A. *et al.* CXCL12 in early mesenchymal progenitors is required for haematopoietic stem-cell maintenance. *Nature* **495**, 227-230 (2013). <https://doi.org/10.1038/nature11926>
- 12 Mendez-Ferrer, S. *et al.* Mesenchymal and haematopoietic stem cells form a unique bone marrow niche. *Nature* **466**, 829-834 (2010). <https://doi.org/10.1038/nature09262>
- 13 Tikhonova, A. N. *et al.* The bone marrow microenvironment at single-cell resolution. *Nature* **569**, 222-228 (2019). <https://doi.org/10.1038/s41586-019-1104-8>
- 14 Seshadri, R., Kutlaca, R. J., Trainor, K., Matthews, C. & Morley, A. A. Mutation rate of normal and malignant human lymphocytes. *Cancer Res* **47**, 407-409 (1987).
- 15 Duesberg, P., Stindl, R. & Hehlmann, R. Explaining the high mutation rates of cancer cells to drug and multidrug resistance by chromosome reassortments that are catalyzed by aneuploidy. *Proc Natl Acad Sci U S A* **97**, 14295-14300 (2000). <https://doi.org/10.1073/pnas.97.26.14295>
- 16 LeBlanc, R. *et al.* Proteasome inhibitor PS-341 inhibits human myeloma cell growth in vivo and prolongs survival in a murine model. *Cancer Res* **62**, 4996-5000 (2002).

- 17 Silva, A. *et al.* An Ex Vivo Platform for the Prediction of Clinical Response in Multiple Myeloma. *Cancer Res* **77**, 3336-3351 (2017). <https://doi.org:10.1158/0008-5472.CAN-17-0502>
- 18 Bravo, R. R. *et al.* Hybrid Automata Library: A flexible platform for hybrid modeling with real-time visualization. *PLoS Comput Biol* **16**, e1007635 (2020). <https://doi.org:10.1371/journal.pcbi.1007635>
- 19 Keats, J. J. *et al.* Clonal competition with alternating dominance in multiple myeloma. *Blood* **120**, 1067-1076 (2012). <https://doi.org:10.1182/blood-2012-01-405985>
- 20 Dutta, A. K., Alberge, J. B., Sklavenitis-Pistofidis, R., Lightbody, E. D., Getz, G. & Ghobrial, I. M. Single-cell profiling of tumour evolution in multiple myeloma - opportunities for precision medicine. *Nat Rev Clin Oncol* **19**, 223-236 (2022). <https://doi.org:10.1038/s41571-021-00593-y>
- 21 Tang, Y. *et al.* TGF-beta1-induced migration of bone mesenchymal stem cells couples bone resorption with formation. *Nat Med* **15**, 757-765 (2009). <https://doi.org:10.1038/nm.1979>
- 22 Hering, S. *et al.* TGFbeta1 and TGFbeta2 mRNA and protein expression in human bone samples. *Exp Clin Endocrinol Diabetes* **109**, 217-226 (2001). <https://doi.org:10.1055/s-2001-15109>
- 23 Martin, T. J. & Sims, N. A. Osteoclast-derived activity in the coupling of bone formation to resorption. *Trends Mol Med* **11**, 76-81 (2005). <https://doi.org:10.1016/j.molmed.2004.12.004>
- 24 Janssens, K., ten Dijke, P., Janssens, S. & Van Hul, W. Transforming growth factor-beta1 to the bone. *Endocr Rev* **26**, 743-774 (2005). <https://doi.org:10.1210/er.2004-0001>
- 25 Wu, M., Chen, G. & Li, Y. P. TGF-beta and BMP signaling in osteoblast, skeletal development, and bone formation, homeostasis and disease. *Bone Res* **4**, 16009 (2016). <https://doi.org:10.1038/boneres.2016.9>
- 26 Edwards, J. R. *et al.* Inhibition of TGF-beta signaling by 1D11 antibody treatment increases bone mass and quality in vivo. *J Bone Miner Res* **25**, 2419-2426 (2010). <https://doi.org:10.1002/jbmr.139>
- 27 Sahbani, K., Cardozo, C. P., Bauman, W. A. & Tawfeek, H. A. Inhibition of TGF-beta Signaling Attenuates Disuse-induced Trabecular Bone Loss After Spinal Cord Injury in Male Mice. *Endocrinology* **163** (2022). <https://doi.org:10.1210/endocr/bqab230>
- 28 Paton-Hough, J. *et al.* Preventing and Repairing Myeloma Bone Disease by Combining Conventional Antiresorptive Treatment With a Bone Anabolic Agent in Murine Models. *J Bone Miner Res* **34**, 783-796 (2019). <https://doi.org:10.1002/jbmr.3606>
- 29 Mohammad, K. S. *et al.* Pharmacologic inhibition of the TGF-beta type I receptor kinase has anabolic and anti-catabolic effects on bone. *PLoS One* **4**, e5275 (2009). <https://doi.org:10.1371/journal.pone.0005275>

REVIEWER COMMENTS

Reviewer #1 (Remarks to the Author):

I acknowledge the tremendous efforts the authors have undertaken to revise the manuscript. I have no further questions.

Reviewer #2 (Remarks to the Author):

The authors have addressed my comments and their response is well integrated in the manuscript.

The only additional feedback I would add is to insert a readme file in the code repository to further ensure reproducibility.

Reviewer #3 (Remarks to the Author):

The authors have answered the questions adequately in the rebuttal, and the revised manuscript has improved. I would accept this manuscript now.

Reviewer #4 (Remarks to the Author): Expert in multiple myeloma in vivo models and imaging

Overall, this is a very interesting model system made by this team, that would be very interesting for the field of myeloma and bone researchers. The rebuttal letter is strong. A few edits remain:

Minor: Fix Fig 5 legend "Ex vivo analysis of histological sections from the U266-GFP myeloma model(N=3-5 tibia per time point.) per time point demonstrates increasing presence of α SMA+ MSCs"

Bortezomib can also induce osteogenesis- is that included in the model equations? I don't think so.

In vivo model for Fig 6 (n,o) is not convincing for the title of the Figure based on the data shown. This also raises questions about the model system and how they are coding for the effects of bisphosphonates in the MM niche overall. In that figure, their data suggest, as others have seen, that Zoledronic acid could be having a direct anti-myeloma effect, independent of the Bortezomib/drug resistance effects. zoledronic acid may modulate the growth and survival of myeloma cells directly. Indeed, studies have demonstrated that a number of bisphosphonates, including zoledronic acid, are able to inhibit myeloma cell proliferation and promote apoptosis in vitro- see the discussion of this paper- by peter croucher- Zoledronic Acid Treatment of 5T2MM-Bearing Mice Inhibits the Development of Myeloma Bone Disease: Evidence for Decreased Osteolysis, Tumor Burden and Angiogenesis, and Increased Survival.

In their in vivo and often in vitro data, the error bars are large and undefined in the legend, and no significance is shown or described on the figure or in the legend. Micrograms should use a symbol instead of just ug. N=4/5 mice is very low for an in vivo study- they could easily show all mice if they did

so few (though a higher n would be ideal). Also, what day are the IVIS images from? (in Fig 6 O it is missing that information.)

Fig 7- in vitro and in vivo data are lacking statistical testing like in Fig 5. In E- total numbers and not just percentages would be good to know as well.

In the rebuttal, they state “Yes, in future iterations we aim to apply multiple known “EMDR” targeting agents such as AMD300, anti-IL6 antibodies and ZOL to name a few, alone or in combination with BTZ, in standard treatment schedules or in adaptive therapy manner. We hope to find the best combination and treatment scheduling to delay the onset of heterogeneous resistant disease.” But I am not convinced the model will indeed give you this information. They also suggest this in your intro to the paper with the line “Further, mathematical models have the potential for clinical translatability and can be used to study the impact of treatment on the tumor microenvironment and delay the onset of resistant disease²⁴⁻³¹.”. Despite this model being interesting, I am not sure that it will be able to give them the data they describe here- so it should not be overstated or it should all be better supported if it is really possible. It would also be good in the model to label what is bone and what is the space around the bone (ie bone marrow?).

They should mention other drawbacks/limitations to the model- not only are no immune cells present, but no adipocytes are present in the model.

Minor issue- the line “Our data demonstrates”. Remember that data are plural. And Check sup Fig 4 legend- the sentence is not complete- “the high resistance probability prevents tumors from going extinct before the acquisition of intrinsic and also drives tumor relapse.” Also, it is not clear what the grey colored pixels represent in that video- in the 3rd panel- since that should be a No EMDR panel, but there are grey pixels that are labeled in the legend as EMDR. Please clarify.

Author's Rebuttal:

We thank the reviewers for their kind words regarding our revised submission. We also greatly appreciate Reviewer #4's additional comments. We have done our utmost to address those points below (in blue text) and by adding new information and data that we believe further strengthens our study.

REVIEWER COMMENTS

Reviewer #1 (Remarks to the Author):

I acknowledge the tremendous efforts the authors have undertaken to revise the manuscript. I have no further questions.

We thank the reviewer for their kind words.

Reviewer #2 (Remarks to the Author):

The authors have addressed my comments and their response is well integrated in the manuscript. The only additional feedback I would add is to insert a readme file in the code repository to further ensure reproducibility.

Thank you. All codes and files supporting our findings will be uploaded to ensure reproducibility. In addition, a readme file has been added to the code repository.

Reviewer #3 (Remarks to the Author):

The authors have answered the questions adequately in the rebuttal, and the revised manuscript has improved. I would accept this manuscript now.

Thank you!

Reviewer #4 (Remarks to the Author): Expert in multiple myeloma in vivo models and imaging

Overall, this is a very interesting model system made by this team, that would be very interesting for the field of myeloma and bone researchers. The rebuttal letter is strong.

A few edits remain:

Minor: Fix Fig 5 legend "Ex vivo analysis of histological sections from the U266-GFP myeloma model (N=3-5 tibia per time point.) per time point demonstrates increasing presence of α SMA+ MSCs"

We apologize for this oversight; we have edited the main text (page 46) to clarify this issue and thank the reviewer for pointing this out.

Bortezomib can also induce osteogenesis- is that included in the model equations? I don't think so.

That is correct. We and others have shown that bortezomib can induce mineralization by osteoblasts (Supp figure 3b). This effect was included in the manuscript (Supplementary Figure 3c), and we demonstrate that when the in-silico model was treated with BTZ, more bone volume was observed (Supp Figure 3F). We have added an additional sentence to the main text (page 9) to help clarify this for readers.

In vivo model for Fig 6 (n,o) is not convincing for the title of the Figure based on the data shown.

This also raises questions about the model system and how they are coding for the effects of bisphosphonates in the MM niche overall. In that figure, their data suggest, as others have seen, that Zoledronic acid could be having a direct anti-myeloma effect, independent of the Bortezomib/drug resistance effects. zoledronic acid may modulate the growth and survival of myeloma cells directly. Indeed, studies have demonstrated that a number of bisphosphonates, including zoledronic acid, are able to inhibit myeloma cell proliferation and promote apoptosis in vitro- see the discussion of this paper- by peter croucher- Zoledronic Acid Treatment of 5T2MM-Bearing Mice Inhibits the Development of Myeloma Bone Disease: Evidence for Decreased Osteolysis, Tumor Burden and Angiogenesis, and Increased Survival.

This is an important point raised by the reviewer. As stated in the discussion by *Croucher et al*¹:

“the mechanisms by which zoledronic acid decreases tumor burden and promotes survival are less clear. Zoledronic acid, by inhibiting bone resorption, may alter the local bone marrow microenvironment, thereby removing important growth and/or survival factors for myeloma cells... Indeed, studies have demonstrated that a number of bisphosphonates, including zoledronic acid, are able to inhibit myeloma cell proliferation and promote apoptosis in vitro. (24-26,50,51) However, in the present study, zoledronic acid did not increase the proportion of apoptotic tumor cells present in the bone marrow, suggesting that the antitumor effect is not mediated by a direct effect on the tumor cells.”

Whilst our own data *in vitro* points to their being a direct effect of Zol on MM cells at very high doses (>10 μ M), previous studies have determined that the bioavailability of zoledronate in the bones of patients who developed osteonecrosis of the jaw and therefore experienced prolonged exposure to zoledronate, to be 0.4 to 5 μ M². At concentrations \leq 10 μ M, we observed no effects on U266 viability *in vitro* (**Supp Fig. 4a and Supp Fig. 9f**). These results are further strengthened by our *in vivo* single agent treatment with Zol, where no significant effect on GFP+ U266 nor RFP+ PSR cell numbers were observed (**Fig. 7e-f and new Supp Fig. 9e**).

To assess whether there was a direct synergistic effect with BTZ and Zol *in vitro* that may account for our observed effects (**Fig 7e**) we have performed additional *in vitro* experiments. Here we show that combining BTZ+ZOL has no added benefit *in vitro* (**new Supp. Fig 9f**). In fact, Zol was mildly antagonistic to BTZ treatment of U266 MM at 8-10 μ M. Similarly, like *Croucher et al*, we conclude that the anti-tumor effect is not mediated by a direct effect on tumor cells but rather by altering the bone marrow microenvironment.

It is a great point raised by the reviewer and we have added additional sentences to explain this observed effect – page 14-15 and also to the discussion.

In their in vivo and often in vitro data, the error bars are large and undefined in the legend, and no significance is shown or described on the figure or in the legend.

We have taken the time to better describe the averages and error bars in the figures. For In vivo experiments, we have changed the figures to show all data points or in some cases add additional supplemental data.

- Updated Figure 4 e-g

- Updated Figure 5 e-h
- Updated Figure 6n
- New Supplementary Figure 9

Micrograms should use a symbol instead of just ug.

Thank you. We have changed ug to μg in the main text, line 1049.

N=4/5 mice is very low for an in vivo study- they could easily show all mice if they did so few (though a higher n would be ideal).

In our experience and hands, the *in vivo* models are very consistent regarding tumor engraftment and the sample size for the in vivo studies have been appropriately powered. With an α of 0.05, 4 mice per group achieves 80% power to detect a 75% reduction in trabecular bone volume (BV/TV) in myeloma bearing mice compared to tumor naïve mice. Similarly, 5 mice per group achieves 90% power to detect the expected bone loss. For these reasons, we chose to use a small, yet appropriate, number of mice for our experiments. We have altered figures 4 and 5 to show all mice in each group. Further, whilst we feel that the summary data for figure 6n better demonstrates the point that ZOL+BTZ decreases sensitive myeloma cell growth, we have added additional spider plots to supplementary figure 9 so that all mice/tumors can be seen.

Also, what day are the IVIS images from? (in Fig 6 O it is missing that information.)

The IVIS images are from day 28, we have included this in the figure legend on page 50 and thank the reviewer for this comment.

Fig 7- in vitro and in vivo data are lacking statistical testing like in Fig 5.

The goal our studies were to use the biological data to validate the in-silico results and show similar effects in each setting rather than to define statistical differences in the biological results. That being said, we have taken the time to perform appropriate statistical tests throughout all figures.

In E- total numbers and not just percentages would be good to know as well.

We have added the raw flow cytometry data to **new supplementary figure 9**

n the rebuttal, they state “Yes, in future iterations we aim to apply multiple known “EMDR” targeting agents such as AMD300, anti-IL6 antibodies and ZOL to name a few, alone or in combination with BTZ, in standard treatment schedules or in adaptive therapy manner. We hope to find the best combination and treatment scheduling to delay the onset of heterogeneous resistant disease.” But I am not convinced the model model will indeed give you this information. They also suggest this in your intro to the paper with the line “Further, mathematical models have the potential for clinical translatability and can be used to study the impact of treatment on the tumor microenvironment and delay the onset of resistant disease²⁴⁻³¹.”. Despite this model being interesting, I am not sure that it will be able to give them the data they describe here- so it should not be overstated or it should all be better supported if it is really possible. It would also be good in the model to label what is bone and what is the space around the bone (ie bone marrow?).

We have taken into consideration these suggestions into our latest version of the manuscript and agree with the reviewer in not overstating the *immediate* clinical translatability of the current version of the

model's results. However, several mathematical models have already been developed to aid in clinical use and to delay resistant disease in patients, most notably in prostate cancer^{3, 4, 5, 6, 7, 8}.

Agent-based models such as the one we describe in our manuscript allow us a small window where we can appreciate the dynamics of tumor-stroma interactions with a high degree of resolution at both the temporal and spatial scales. But models that capture the clinical scale of the disease need to also consider tumor burden with much larger populations over multiple and possibly heterogeneous microenvironments. Our current work could be seen as an early stage in the process to make this disease and the ecological dynamics that influence its evolution, amenable to integrated approaches combining mathematical and experimental models with robust clinical data that could have clinical impact. We have altered the manuscript accordingly (introduction page 6 and discussion page 23)

They should mention other drawbacks/limitations to the model- not only are no immune cells present, but no adipocytes are present in the model.

We have updated our manuscript to include limitations such as not including other drivers of myeloma such as adipocytes, immune cells and their derived factors (page 21).

Minor issue- the line "Our data demonstrates". Remember that data are plural.

Thank you. We have corrected this oversight in the main text, line 36 and 42.

And Check sup Fig 4 legend- the sentence is not complete- "the high resistance probability prevents tumors from going extinct before the acquisition of intrinsic and also drives tumor relapse."

We thank the reviewer for noticing this error, we have fixed the sentence in question (supplementary page 40)

Also, it is not clear what the grey colored pixels represent in that video- in the 3rd panel- since that should be a No EMDR panel, but there are grey pixels that are labeled in the legend as EMDR. Please clarify.

We apologize for the oversight; these cells are resistant cells that are in close proximity to MSCs or BDFs but they are not receiving EMDR as they already have intrinsic resistance to BTZ treatment. We have adjusted the legend appropriately for figure 7, page 55.

References Cited:

1. Croucher PJ, *et al.* Zoledronic acid treatment of 5T2MM-bearing mice inhibits the development of myeloma bone disease: evidence for decreased osteolysis, tumor burden and angiogenesis, and increased survival. *J Bone Miner Res* **18**, 482-492 (2003).
2. Scheper MA, *et al.* A novel bioassay model to determine clinically significant bisphosphonate levels. *Support Care Cancer* **17**, 1553-1557 (2009).
3. Brady-Nicholls R, *et al.* Predicting patient-specific response to adaptive therapy in metastatic castration-resistant prostate cancer using prostate-specific antigen dynamics. *Neoplasia* **23**, 851-858 (2021).

4. Pasetto S, Enderling H, Gatenby RA, Brady-Nicholls R. Intermittent Hormone Therapy Models Analysis and Bayesian Model Comparison for Prostate Cancer. *Bull Math Biol* **84**, 2 (2021).
5. West JB, Dinh MN, Brown JS, Zhang J, Anderson AR, Gatenby RA. Multidrug Cancer Therapy in Metastatic Castrate-Resistant Prostate Cancer: An Evolution-Based Strategy. *Clin Cancer Res* **25**, 4413-4421 (2019).
6. Gallaher J, *et al.* Improving treatment strategies for patients with metastatic castrate resistant prostate cancer through personalized computational modeling. *Clin Exp Metastasis* **31**, 991-999 (2014).
7. Macklin P, Edgerton ME, Thompson AM, Cristini V. Patient-calibrated agent-based modelling of ductal carcinoma in situ (DCIS): from microscopic measurements to macroscopic predictions of clinical progression. *J Theor Biol* **301**, 122-140 (2012).
8. Alfonso JCL, *et al.* Tumor-immune ecosystem dynamics define an individual Radiation Immune Score to predict pan-cancer radiocurability. *Neoplasia* **23**, 1110-1122 (2021).

REVIEWERS' COMMENTS

Reviewer #4 (Remarks to the Author):

This is really excellent work.